# Transition from simple to complex contagion in collective decision-making

Nikolaj Horsevad [1]✉, David Mateo [2], Robert E. Kooij[3,4], Alain Barrat [5,6] & Roland Bouffanais [1]✉

How does the spread of behavior affect consensus-based collective decision-making among animals, humans or swarming robots? In prior research, such propagation of behavior on social networks has been found to exhibit a transition from simple contagion—i.e, based on pairwise interactions—to a complex one—i.e., involving social influence and reinforcement. However, this rich phenomenology appears so far limited to threshold-based decision-making processes with binary options. Here, we show theoretically, and experimentally with a multi-robot system, that such a transition from simple to complex contagion can also be observed in an archetypal model of distributed decision-making devoid of any thresholds or non-linearities. Specifically, we uncover two key results: the nature of the contagion—simple or complex—is tightly related to the intrinsic pace of the behavior that is spreading, and the network topology strongly influences the effectiveness of the behavioral transmission in ways that are reminiscent of threshold-based models. These results offer new directions for the empirical exploration of behavioral contagions in groups, and have significant ramifications for the design of cooperative and networked robot systems.

[1] University of Ottawa, Ottawa, Canada. [2] Kido Dynamics, Lausanne, Switzerland. [3] Delft University of Technology, Delft, The Netherlands. [4] The Netherlands Organization for Applied Scientific Research (TNO), The Hague, The Netherlands. [5] Aix Marseille Univ, Université de Toulon, CNRS, CPT, Turing Center for Living Systems, Marseille, France. [6] Tokyo Tech World Research Hub Initiative (WRHI), Tokyo Institute of Technology, Tokyo, Japan. ✉email: nikolajhorsevad@gmail.com; roland.bouffanais@uottawa.ca

Complex systems, be them natural or artificial, operate on the basis of particular connectivity between constituting elements, which orchestrates the execution of specific dynamical processes. Such a high-level abstraction encompasses wildly different systems giving rise to a range of emergent behaviors, such as fish schooling[1–3], social opinion formation[4], disease spreading[5], cascading failures in power grids[6,7], target tracking by swarm robotic systems[8,9], etc. Such collective dynamics have been found to be crucially dependent on the underlying network topology[2,3,5,10–15], which conditions the efficient transmission of behavioral change.

The propagation of state changes within a social system—or an engineered networked one—has been acknowledged to be akin to a contagion process, which can be either "simple" or "complex"[13,16–19]. With a simple contagion, the behavior propagates through a single exposure or interaction. On the other hand, if social reinforcement is required—following Centola and Macy's definition[13]: "if its transmission requires an individual to have contact with two or more sources of activation"—the contagion is said to be complex.

Numerous models of behavioral propagation in networked systems (including social ones) have been considered over the years. Threshold models have been the predominant modeling framework used to characterize a wide range of complex contagion processes, such as the adoption of technological innovations[4] or preventative health measures[20], and the spread of misinformation on social media[19,21] (Note that complex-like contagions are also observed with stochastic models of epidemic-like processes[22–25]). The growing interest in threshold models can be traced to their mathematical simplicity, their paradigmatic nature and their success in modeling the spread of behaviors in various social settings[4,13,16,26]. These deterministic models assume that agents can be in two states (inert or activated), and that a particular agent becomes activated if a fraction of its neighbors (in the network sense) larger than a given threshold are themselves activated[17,27]. These threshold models perfectly fit Centola & Macy's original definition of a complex contagion, whereby a transition from simple to complex contagion takes place when increasing the threshold beyond a value corresponding to having more than one activated neighbor[13,16,17]. However, the key concept of activation may not be as straightforward when considering models lacking a threshold, which can become an issue when trying to use the existing definition of a complex contagion.

Here, we show theoretically and experimentally that such a transition from simple to complex contagion, as originally identified in threshold-based models, can also be exhibited by another general class of collective decision-making processes; specifically, a class of models based on consensus and devoid of any thresholds or nonlinearities. Using a new way of characterizing complex contagions, we uncover their existence in consensus-based dynamics. Specifically, we shed a new light on some fundamental mechanisms underpinning networked systems, which may support the study of a vast range of collective behaviors in both the animal and social worlds.

Unsurprisingly, the network topology plays a pivotal role in this study[2,3,10–13,15,16]. It is known that it strongly affects both spreading types in threshold models, albeit in fundamentally different ways. While simple contagions are enhanced by short network distances[5], complex ones are amplified by high levels of clustering[4,5,14,16]. Within the framework of threshold models, the behavior or state being transmitted is of a binary nature: "active" or "inactive". This simplification clearly facilitates the tracking of behavioral cascades from a source (or multiple sources) to the entire system (or parts of the system). This feature serves well the purpose of studying collective decision-making processes involving two options, such as voting, adoption of innovations, binary opinion dynamics[4,20]. However, numerous collective decisions are more complex and involve a continuum of options rather than just a binary set[28–30]. A full understanding of the influence of network metrics on consensus-based decision-making involving behavioral propagation and/or external perturbations to the consensus is lacking. Such knowledge would help gain insight into the disturbed collective dynamics of social and animal groups, e.g., when responding to a predator's attack or to misleading information on social networks.

Biologists have indeed recently acknowledged the profound similarities between human and animal social behaviors. For instance, Sosna et al. recently reported a study on the "fear response" of a school of fish collectively making fast decisions under risky conditions[3]. They found that the properties of the network (their "social connectivity") are the primary factors responsible for the high collective responsiveness of the school in terms of the number of behavioral cascades and their sizes. Furthermore, Firth[18] has made the case that complex contagions might be key to explaining some specific collective animal dynamics, especially those with socially transmittable behaviors. In the case of direct behavioral transmission in mobile animal groups—schooling fish, flocking birds, swarming insects[31–34]—a full understanding of the nature of the propagation is still lacking[2,3]. The markedly fast spread of behaviors within animal groups—such as waves of response, evasive maneuvers in schools of fish[2,3,11,35], and collective turns in flocks of starlings[36]—has been a source of inquiry for a long time[35]. Recent large-scale empirical evidence with fish and birds have revealed the intricate patterns of interaction among individuals[2,3,36–39], which underpin the behavioral cascades throughout the group. A key element to these inter-agent interactions is alignment—metric or topological—that introduces a consensus component to the collective decision-making process. Unlike binary threshold-based models, such orientation consensus-based ones do not lend themselves well to the tracking of behavioral cascades given the nonbinary nature of the state variable. Moreover, as with all ethological results, even if it is possible to modify the interaction among agents in some ways[3], it is virtually impossible to fully control all aspects of it.

However, biologists have started using robotic agents in place of animals to be able to measure and quantify some features of interest[40,41]. Hence, by following a similar approach with a multirobot system, one could compare the effectiveness of the social transmission of information when changing the local interaction rule, i.e., when changing the topology of the interaction network.

Here, we specifically consider a collective decision-making process reminiscent of a group escape response. Using the leader–follower consensus (LFC) dynamics—a particular instance of the general control-theoretic framework of the Taylor model[30,42,43]—we are able to study the behavioral contagion within a group of networked agents driven by a single leader acting as a stimulus of tunable frequency[11,12]. With this linear-time-invariant (LTI) dynamics, by varying the network topology—specifically the clustering coefficient, average shortest path, and Kirchhoff index—we observe that slow-paced (resp. fast-paced) stimuli propagate in ways reminiscent of a simple (resp. complex) contagion. Furthermore, we uncover a transition from simple to complex contagion when varying the pace of the stimulus—i.e, its frequency. This transition is made apparent by measuring the effectiveness of the behavioral propagation—quantified here by means of the collective frequency response[12]—when varying the topological features of the interaction networks (e.g., clustering coefficient, average shortest path, etc.). In addition, using a robotic experimental test-bed comprising ten networked agents performing an angular heading consensus similar to those

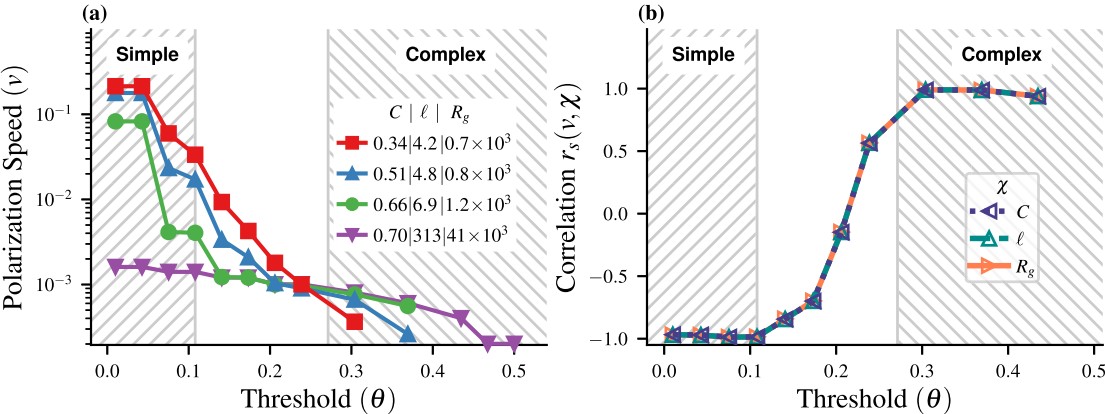

**Fig. 1 Linear threshold model on WS networks of $N = 10,000$ nodes, with fixed average degree $\langle k \rangle = 16$ and uniform threshold $\theta$.** Initially, a single randomly selected seed node and its neighbors are activated. **a** Average polarization speed $v$ when the cascade size is 30% active nodes on the network, with lines stopping prematurely if the fraction is never reached, shown for a representative set of network metrics (Supplementary Fig. S1 for other cascade sizes). The WS rewiring probability $p$ is used to generate network samples having specific values of $\chi \in \{C, \ell, R_g\}$. **b** Spearman's correlation coefficients $r_s$ between the polarization speed $v$ and each network property $\chi \in \{C, \ell, R_g\}$.

observed in collective turns of flocking starlings[39], we unambiguously confirm that the behavioral propagation has the features of a complex contagion when driven by fast-paced stimuli.

These results have far-reaching implications for several reasons. First, they extend the concept of transition from simple to complex contagion—heretofore limited to binary threshold-based models[16,17]—to the continuous class of consensus-based models. It is worth highlighting that the original linear threshold model (LTM) with binary options has been extended to a continuous threshold model (CTM) of cascade dynamics, which involves nonlinearities and a threshold[44]. However, no evidence of a transition from simple to complex contagion has been reported for the CTM. Second, these results reveal that the nature of the contagion—simple or complex—is directly related to the type of behavior spreading, and more specifically to the pace of its intrinsic dynamics—e.g., slow external perturbations vs. collective startle response. Lastly, the insights gained from this study could offer new directions for biologists and social scientists to explore and experiment with animal and human groups respectively. They could also be harnessed to improve the design and robustness of engineered networked systems (e.g., internet of things, sensor networks, swarm robotics).

## Results

**Polarization speed in threshold models.** Complex contagions have been originally uncovered and studied using the archetypal LTM[13,16,17,27], where nodes become active when the fraction of their active neighbors crosses a certain threshold $\theta$. Specifically, the LTM dictates that for any agent $i$, the binary state variable $s_i(t)$ is updated according to

$$s_i(t+1) = \begin{cases} 1 & \text{if } \langle s_j(t) \rangle_{j \sim i} > \theta, \\ 0 & \text{otherwise,} \end{cases} \quad (1)$$

where $\langle \cdot \rangle_{j \sim i}$ is the average over all neighbors of $i$ ("Methods"). Significant attention has been dedicated to understanding the interplay between $\theta$ and network topology for global cascades to occur (i.e., yielding an activation of 99% of nodes)[13]. It has been repeatedly reported that complex contagions spread "faster and further"[16,20] on highly clustered networks, as compared to simple contagions[13,16,45]. It is worth stressing that the investigation of this important statement remains limited to the long-termdynamics of global cascades[46]. On the other hand, the early dynamics of contagion affecting a smaller fraction of

nodes—say cascades of 30% activated nodes, which is still macroscopic—has been relatively overlooked. There has been no attempt to relate the actual speed of contagion with the transition from simple to complex contagion for incomplete—i.e., nonglobal—cascades. This speed of contagion, also known as diffusion speed, is the number of infected nodes per unit time and is increasingly recognized as an important indicator of the contagion dynamics[11,46–49].

A useful metric to analyze this is given by the so-called polarization speed $v = (P(t) - P(0))/t$ at instant $t$, where $P(t)$ measures the polarization of the system ("Methods"). This quantity gives an indication of the speed at which a random activator node and its neighbors can activate a given fraction of nodes (here 30%; "Methods" and Supplementary Fig. S1 for other fractions)[11,17,22,27]. Here, we consider this polarization speed, which is identical to the average speed of diffusion considered in ref. [46] and is also somehow related to the concepts of spreading speed, half prevalence time, or time until half the network is infected[48,50,51], and we study how it relates to some network descriptors used for the study of the spread of misinformation in temporal network epidemiology[52].

Following in the footsteps of Centola & Macy, we use their version of the LTM[16] to study the transition from simple to complex contagion. Simple contagions are observed in LTMs with low thresholds $\theta$, which lead to an increased spreading rate on networks with smaller values of the average shortest path $\ell$[5,45]. As $\theta$ increases, the contagion becomes complex and for the spreading process to endure, the network must possess a sufficiently high value of the clustering coefficient $C$[13,17]. Here, we consider the classical small-world Watts–Strogatz (WS) networks[5], which are constructed on the basis of a single free parameter, namely the rewiring probability $p$. By varying $p$, we can effectively tune $C$, $\ell$, or the Kirchhoff index $R_g$ of the network ("Methods"). The Kirchhoff index $R_g = N \sum_{i=2}^{N} \lambda_i^{-1}$, is a distance metric based on the eigenvalues $0 = \lambda_1 < \lambda_2 < \cdots < \lambda_N$ of the Laplacian matrix[53] ("Methods").

The variations of the polarization speed $v$ with $\theta$ are reported in Fig. 1. At a low threshold, the polarization speed increases when $\ell$ decreases (and $C$ decreases as well), which is characteristic of simple contagions (Fig. 1a). At higher $\theta$ values, this trend is inverted and we retrieve the well-known complex contagion phenomenology in which $v$ increases with $C$ (and also with $\ell$). These trends can also be appreciated by observing the particular network topology corresponding to the purple curve (high $C$, $\ell$

and $R_g$) in Fig. 1a. It goes from producing the worst-performing simple contagion (out of the four topologies considered here) at low $\theta$, to generating the best performing complex contagion at higher threshold values. As expected, the transition region corresponds to intermediate values of the threshold such that the ordering of the different networks does not reveal an unambiguous simple or complex contagion. It is worth noting that highly clustered networks can sustain a complex contagion at higher thresholds compared to the more rewired networks, which lack ample clustering to sustain spreading[13].

To analyze this transition, we calculate Spearman's correlation coefficient $r_s$ between the polarization speed $v$ and each network metric $\chi \in \{C, \ell, R_g\}$, for each threshold value $\theta$ ("Methods" and Supplementary Fig. S2). There is a marked transition from $r_s \approx -1$ for $\theta < 0.1$, to $r_s \approx +1$ when $\theta > 0.28$ (Fig. 1b). These thresholds mark the cutoff of purely simple and complex contagion—used to draw the shaded regions in Fig. 1a—with the transition region in between experiencing a more complex interplay of the network parameters and the resulting polarization speed. Note that the correlation between the three network parameters and the polarization speed are extremely similar, owing to the well-known fact that the WS network model has only one free parameter (the rewiring probability $p$; see ref. [5]). These results are in good agreement with other observations of this transition[5,16,17,45], thereby suggesting that $v$, characteristic of the early contagion dynamics, is both an adequate and effective indicator of the nature—simple or complex—of the behavioral propagation.

**Transition from simple to complex contagion in consensus models**. In the previous section, we showed that with the classical LTM, the transition from simple to complex contagion can be analyzed and understood from a new angle—the speed of contagion—using $v$ as an indicator of such speed. The question now is whether such a transition can be observed with other collective decision-making processes that do not involve binary state variables with cascades of changes, nor nonlinear mechanisms/thresholds. Specifically, we consider the canonical linear time-invariant Taylor model[30,42,43], which has been widely used to characterize a vast breadth of collective behaviors[1,11,37,38,54,55] and decision-making[30,56]. In the Taylor model, the agents (nodes) seek to reach a consensus by performing some average of their own state along with those of their neighbors in the network sense. However, flocking birds and schooling fish in the wild seldom reach a complete consensus given their incessant collective maneuvering: the convergence to a stationary state clearly does not apply to their dynamics. This is even more true when these animal groups are dealing with predator attacks or other external perturbations. Indeed, accumulating empirical evidence shows that swift behavioral propagation is the true hallmark of collective behavior, rather than high consensus or polarization[37]. It can therefore be said that although consensus is at the root of their collective actions, these systems effectively tend to operate away from consensus[11,12].

Considering the particular Taylor model corresponding to the LFC dynamics, one can drive the system away from consensus by imposing a given dynamics to the "leader" (also known as "stubborn", "zealot", "informed" agent in some contexts[30]). The leader's behavior then propagates to the neighboring agents, and further to the entire system, thereby determining the emergent collective response. From the control-theoretic perspective, this leader introduces a time-varying input signal into the system. However, this behavioral propagation intricately depends on the network topology, as well as on the leader–follower consensus dynamics considered. Here, we consider $N$ agents with state

variable $x_i(t)$ seeking to follow the arbitrary trajectory $x_0(t) = u(t) = \sin \omega t$ of the leader agent $i = 0$, by means of the following linear distributed consensus:

$$\frac{dx_i}{dt} = \sum_{j=1}^{N} w_{ij} x_j(t) + w_{i0} u(t), \qquad (2)$$

where $w_{ij}$ is a weight related to the interaction between agents $i$ and $j$ ("Methods"). The collective frequency response of the system, $H^2(\omega)$ (Eq. (7) in "Methods"), can be interpreted as the number of agents that are able to respond or follow the leader's behavior, as a function of its frequency $\omega$[11,57].

The LFC dynamics at low frequency ($\omega \to 0$) has been comprehensively studied. For instance, it is well known that the collective response increases as $\ell$ decreases[14,43,58]. This phenomenology is analogous to that of a simple contagion (cf. the increase of the polarization speed $v$ as $\ell$ decreases for the LTM at low threshold $\theta$, Fig. 1a). Given the transition from a simple to a complex contagion when increasing $\theta$ in the LTM, one is naturally led to consider the possible existence of a transition in the LFC when increasing the frequency $\omega$ of the leader's dynamics.

To investigate if the LFC indeed exhibits such a transition, we follow the same approach as for the LTM. We analyze the collective response on networked systems having 240 nodes with a fixed average degree of $\langle k \rangle = 16$. Using the same family of small-world WS networks[5] ("Methods"), we are able to compute analytically $H^2(\omega)$ (Fig. 2a) for the same values of the clustering $C$ as the ones previously used for the LTM (Fig. 1). It is worth adding that similar results are obtained with a family of scale-free networks (Supplementary Fig. S4). Unsurprisingly, at low frequency ($\omega \lesssim 10^{-2}$), we observe a phenomenology consistent with a simple contagion, namely $H^2$ increases as $\ell$ decreases. Upon increasing $\omega$, this trend is reversed and $H^2$ grows with $C$ in ways that are reminiscent of a complex contagion. However, to ascertain that this phenomenology is indeed a transition from a simple to a complex contagion, one has to verify that the simple contagion at low frequency is driven by $\ell$ or $R_g$, while the complex one at high frequency is controlled by $C$. To this aim, we calculate the Spearman's correlation coefficient $r_s$ between $H^2$ and $\chi \in \{C, \ell, R_g\}$ (Fig. 2b). For these three network metrics, $r_s$ exhibits a clear sigmoidal trend from $-1$ at low frequency to $+1$ at high frequency. This trend echoes the one observed with the LTM when varying $\theta$ (Fig. 1b), with a transition region in the middle. Let us note that the important element here is the presence of a transition regardless of the actual values of the upper (resp. lower) bound of the simple (resp. complex) contagion region. Although beyond the scope of this study, a thorough analysis of the profound nature of this transition—e.g., cross-over, phase transition—might help in systematically defining the extent of this transition region.

However, we are still unable to conclude that $C$ is fully responsible for the observed trend at high frequency with the LFC, although this fact is well known for the LTM at high $\theta$. To reach this conclusion, we have to address a well-known structural constraint with the WS networks, namely the fact that they are constructed by means of a single parameter—the rewiring probability $p$[5,59]. This is clearly visible in the insert of Fig. 2b, where $C$ monotonically increases with $R_g$ and $\ell$ (see Supplementary Fig. S9). To overcome this issue, we include additional WS networks—with different values of the average degree—and select a subsample of these networks having uncorrelated network metrics ("Methods" and insert of Fig. 2c). Given this extended network sampling, we need to account for the effects of degree variations[12,53,60], and as such we impose a normalization procedure (overbar notation) for all quantities of interest: $\bar{H}^2, \bar{C}, \bar{\ell}$

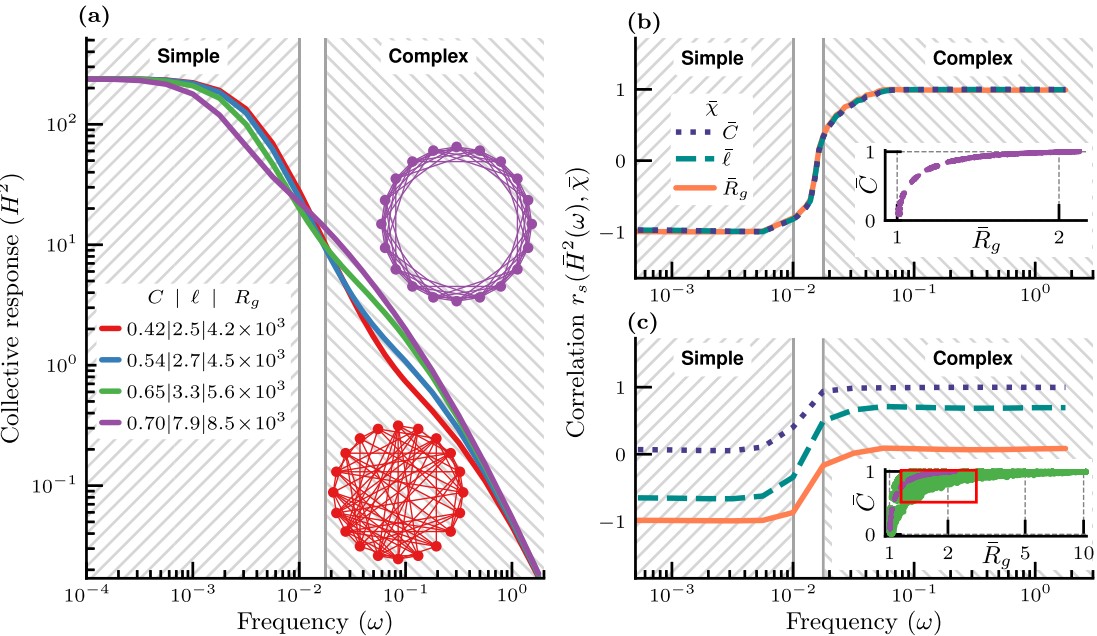

**Fig. 2 Leader–follower consensus model on WS networks of $N = 240$ nodes.** The collective response is averaged over all nodes as the leader. **a** Collective response $H^2$, for a subset of the networks with average degree $\langle k \rangle = 16$; **b, c** Spearman's correlation coefficient $r_s$ between the collective response $\bar{H}^2$ and normalized network metrics $\bar{\chi} \in \{\bar{C}, \bar{\ell}, \bar{R}_g\}$. **b** Networks with average degree $\langle k \rangle = 16$. **c** Networks with degrees $\langle k \rangle \in \{4, 6, ..., 32\}$, limited to a range of uncorrelated network parameters. Note that the inserts show the distribution of the normalized clustering coefficients $\bar{C}$ and Kirchhoff indexes $\bar{R}_g$ used to compute $r_s$ (Supplementary Figs. S5–S6). The purple points in the inserts are for networks with $\langle k \rangle = 16$, while those in green in panel **c** are for other values of $\langle k \rangle$. The red box in the insert of panel **c** highlights the dots corresponding to the subsample of networks used to generate the Spearman's correlation coefficient in panel **c**, with $r_s(\bar{C}, \bar{R}_g) \approx 0.03$ (Supplementary Fig. S3 for other possible subsampling of networks).

and $\bar{R}_g$ ("Methods"). Using this methodology, we obtain the following key result: at low frequency, $\bar{H}^2$ is highly (negatively) correlated with $\bar{R}_g$ and practically uncorrelated with $\bar{C}$, while the opposite is true at high frequency. This unambiguously confirms our hypothesis that the consensus-based behavioral propagation at low (resp. high) frequency is of the simple (resp. complex) contagion type.

It is worth emphasizing that the only commonality between the LTM and the LFC is that both constitute a collective decision-making protocol exhibiting a transition from simple to complex contagion. Although there are obvious similarities between the phenomenology of this transition in both cases, as illustrated by the behavior respectively of the polarization speed (Fig. 1) and of the collective response (Fig. 2), we do not seek here to establish any formal equivalence between their respective control parameters ($\theta$ for the LTM and $\omega$ for the LFC).

**Complex contagion with networked robots governed by non-linear heading consensus.** The theoretical result described above, concerning the canonical LFC, is compelling in several ways. First, it reveals the existence of a transition from simple to complex contagion in the absence of threshold-based mechanisms. This transition in the behavior of consensus-based decision-making processes occurs when varying the inherent pace of the behavior that is propagated through the system. Second, it has significant ramifications for the understanding of some collective behaviors subjected to external fast-pace perturbations, e.g., following a predator's attack leading to collective evasive maneuvers by schools of fish or flocks of birds. As recently stressed by Firth[18], the profound nature of these socially transmittable behaviors has yet to be fully understood. Firth makes clear that the concept of complex contagion has been relatively overlooked by biologists, although it might help explain a vast breadth of

collective animal behaviors[18]. In addition, there has been emerging evidence of complex contagion in some schooling behaviors of golden shiners exhibiting highly clustered interaction networks[2]. As already mentioned, dealing with wild animal groups is not only challenging from the practical standpoint, but it also restricts the ability to analyze the influence of the deeply-ingrained nature of the interaction among agents.

The use of robotic systems—in lieu of biological ones—has been considered to overcome some of these challenges and to offer a new toolkit to deepen our understanding of collective animal behaviors[40,41]. Although simulations offer unique ways to systematically analyze algorithms of collective dynamics, they inevitably reduce the fidelity of the model to achieve computational tractability. Indeed, the simulation-reality gap in robotics is known to be exacerbated with multirobot systems[61]. For instance, simulations can fail to adequately capture: (1) the complex physical interactions among agents, (2) the inherent variability among units forming the group, and (3) the fine details of real-world settings in which the agents are embedded. As a matter of fact, achieving high-fidelity simulations often requires the input or feedback from physical experiments.

Data gathered in robotico, with a highly controllable and controlled environment, enable a rapid investigation of a number of hypotheses about collective behaviors. In turn, the outcome of such investigations can serve biologists to identify new directions to explore, test and validate with empirical data. Following this strategy, we use a networked robotic system to assess various socially transmittable behaviors when changing the topology of the interaction network in the presence of a collective decision-making with a nonlinear component. It is worth stressing that the results obtained with the LFC (Fig. 2) are for a linear system dynamics, yet they reveal a surprisingly complex phenomenology. Nonetheless, some collective decision-making processes among moving animals are based on a consensus associated with the

direction of travel, and are inherently nonlinear. Such nonlinear interactions among conspecifics have recently been shown to be responsible for sudden directional switches in groups of pigeons[62].

Therefore, we carried out a series of experiments on nonlinear leader–follower heading consensus with a collective of ground robots where each one aligns its direction of travel with that of their neighbors in the network sense ("Methods"). As a consequence, changing the structure of the interaction network —and in particular its properties such as $\chi \in \{C, \ell, R_g\}$— modifies the nature of the neighborhoods involved in the nonlinear heading consensus. This type of collective decision-making—closely related to Vicsek's canonical model of collective motion[63]—bears some resemblance with the Taylor model analyzed previously. However, a physical embodiment of such a complex system involves significant deviations from the ideal scenario corresponding to the theoretical calculations obtained for the LFC. For instance, while the dynamics of the robots are ultimately governed by physical processes that are continuous in time, the units sense each other's state using asynchronous, discrete communications with stochastic delays and communications dropouts[12].

The networked robotic system comprises ten robots (one leader and nine followers, "Methods") equipped with a "swarm-enabling" unit—providing on-board data-processing, computing and distributed communication capabilities—that allows the system to perform a collection of decentralized cooperative control strategies[64]. The leader continuously rotates at a fixed frequency $\omega$ in the range $0.03\,\text{Hz} < \omega \leq 0.3\,\text{Hz}$. Each robot, including the leader, periodically transmits its heading information to its direct neighbors as per the specific network topology considered. Three distinct topologies ("Random", "Ring", and "Caveman" as shown at the bottom of Fig. 3) are selected on the basis that they have the same average degree $\langle k \rangle = 4$, yet notably different clustering coefficients $C$ ("Methods").

It is worth highlighting that this experimental setup and methodology are identical to the one reported in ref. [12], except for the fact that different network topologies are considered here. According to the analysis of the LFC, the collective response of this networked robotic system is expected to go down when increasing the frequency $\omega$ of the leader agent. This appears clearly in Fig. 3a for all three networks considered. The decrease in the collective response is essentially the same at low frequency ($\omega \leq 0.06\,\text{Hz}$) for all cases. Above that frequency however, we observe a strong difference between all three topologies in the

range $0.06\,\text{Hz} \leq \omega \leq 0.2\,\text{Hz}$. This intermediate range of frequencies shows a rich collective behavior since the dynamics of the leader is rather "fast" thereby preventing any form of heading consensus to be achieved. This places us in a regime similar to that of animal groups dealing with fast-paced perturbations. Beyond 0.2 Hz, the highest collective response, achieved with the "Caveman" network, experiences a sharper decline, which we suspect would go down to the same low level as for the "Random" and "Ring" networks at higher frequencies. Unfortunately, increasing the frequency above 0.3 Hz is not possible in practice due to a limit in the achievable rotation frequency of the leader.

To compensate for this experimental limitation, we perform simulations of the leader–follower nonlinear heading consensus dynamics with $9 + 1$ agents by integrating the system of Eq. (14) (Fig. 3b). Unsurprisingly, these simulations show higher levels of collective response at low frequency, compared to the experimental ones, as they correspond to an idealized communication between agents. In addition, the frequency at which the differences between the different networks become apparent is higher in the simulation case with respect to the experimental one. Finally, we can reach a higher frequency in the simulations than in the experiments, and the results do show the expected merging of the three collective responses at the highest frequency values (that were not reached in Fig. 3a).

Following the analytical results with the LFC (Fig. 2), we are able to further analyze the robotic experiments. The complex contagion induced by the leader unit to the nine followers is clearly visible in the intermediate frequency range $0.06\,\text{Hz} \leq \omega \leq 0.2\,\text{Hz}$. The results are indeed in excellent agreement with the complex contagion phenomenology studied previously in much larger networks ("complex" hatched region of Fig. 2a). When the robotic units are interconnected by means of the most clustered network ("Caveman"), the behavioral spread is the most effective and the collective response the highest. On the other hand, when decreasing the clustering between robotic units—from "Caveman" to "Ring" and ultimately to "Random"—the effectiveness of the spread of the leader's behavior is reduced, and so is the ensuing collective response at the group level.

We note on the other hand that, given that the experiment is limited to ten robotic units with an average degree $\langle k \rangle = 4$, the diameters and Kirchhoff indices for all three networks are small. Therefore, the simple contagion process at low frequency ($\omega \leq 0.06\,\text{Hz}$) is simply too rapid and no meaningful distinction

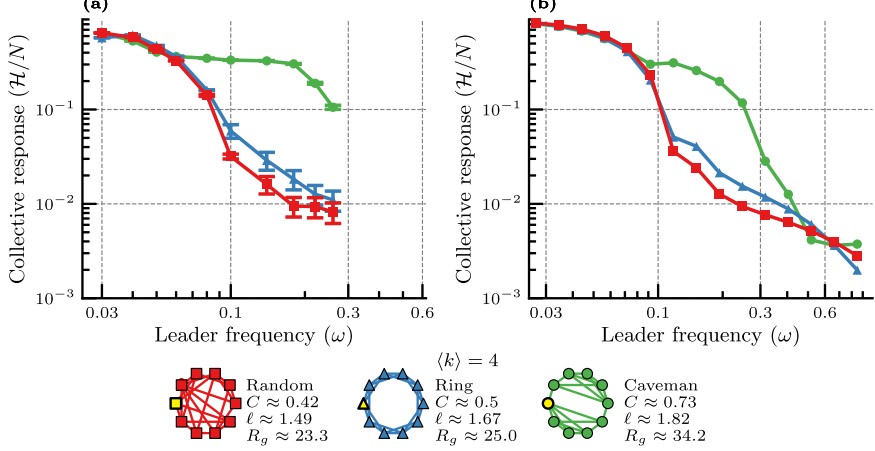

**Fig. 3 Experimental analysis with a leader–follower networked robotic system networked by means of three topologies: "Random", "Ring", and "Caveman" with $\langle k \rangle = 4$ and $9 + 1$ agents ("Methods").** The graphs represent the network topologies used in the series of experiments, with the selected leader node depicted in yellow: **a** Experimental results of angular consensus dynamics (mean value and associated standard deviation; "Methods"), **b** simulation results of the nonlinear angular consensus dynamics.

between the considered topologies can be observed experimentally (Fig. 3a).

## Discussion

In summary, we have shown theoretically and experimentally that complex contagions are more general and prevalent than originally thought, and that transitions from simple to complex contagion are not limited to threshold-based models. Instead, our results suggest that such a transition might be a general feature in some classes of collective decision-making processes.

Network science has been instrumental in uncovering the existence of complex contagions by Centola & Macy[13,16] in the social sciences. Since these seminal works, complex contagions have been observed in other spreading dynamics over complex networks, including epidemic-like stochastic models[22–25]. However, and as already mentioned, the study of the transition from simple to complex contagion has still been limited to the original LTM used by Centola & Macy, as well as social network experiments with binary options in decision-making. Hence, our discovery of this very transition within a fully linear decision-making protocol devoid of any thresholds greatly expands the relevance of this phenomenon to a vast breadth of collective decision-making processes beyond the social sciences, and including collective animal behavior and collective robotics.

At this point, it is critical to realize that ascertaining the exact nature of a contagion—simple or complex—remains challenging. Because of that, some scientists still cautiously use the term "complex contagion" or even refer to it obliquely: e.g., "complex contagion flavor"[65]. Even with the full details of: (1) a given collective dynamics—e.g, linear threshold dynamics for a given value of $\theta$, or a leader–follower consensus dynamics for a given value of $\omega$—as well as (2) the interaction network topology among agents, one is lacking a definite way of characterizing the behavioral contagion. This work offers a key conceptual advance in overcoming this challenge. Indeed, the network-based classification proposes to vary the topology with the goal of either increasing or reducing both the clustering coefficient $C$ and the Kirchhoff index $R_g$. In the event that the effectiveness of the behavioral propagation is found to be positively correlated with $C$ and uncorrelated with $R_g$, then the contagion is complex. Conversely, if a behavioral propagation is negatively correlated with $R_g$ and uncorrelated with $C$, then it can be classified as a simple contagion (Fig. 2c). It is therefore worth stressing that this complex networks characterization of simple/complex contagions is agnostic to the actual details of the collective decision-making process. In all cases investigated here, the simple or complex nature of a behavioral propagation is determined by the value of a control parameter of the dynamical process taking place at the node level: e.g., the threshold $\theta$ for the LTM and the inherent pace of the behavior $\omega$ for the LFC. The proposed characterization allows us to identify the intervals of the control parameter associated with simple or complex contagions, with a transition region in between (Figs. 1 and 2).

Despite the apparent relative simplicity of this complex networks characterization of contagions, one should not underestimate the associated practical challenges for scientists studying a specific collective decision-making dynamics among humans, within animal groups, or with networked robots. Varying the network topology might not always be achievable as previously stressed in the case of flocks of birds or schools of fish. This approach was central to Centola's study of humans involved in an online social experiment, in which the topology of the social network was controlled and manipulated[20]. This groundbreaking experiment has been made possible thanks to the emergence of online social networks and related technology. Clearly, such an approach could not be considered and implemented with animal groups. However, as our networked robotic experiment shows, the use of artificial agents mimicking animal behaviors offers a new toolkit to analyze and gain insight into collective animal behavior. With this approach, one can carry out the proposed complex networks characterization and ascertain the nature—simple or complex—of the behavioral propagation. Last but not least, one can also modify the triggering behavior—in our experiment, the stimulus associated with the frequency of the leader—to possibly uncover a transition from simple to complex contagion. Such an approach opens new doors toward understanding the mechanisms underpinning such collective behaviors.

Given how ubiquitous collective decision-making is in human societies, animal groups, and networked multiagent systems, these new results will have profound ramifications for our understanding of numerous phenomena in these fields. The importance of these results goes beyond the class of consensus-based protocols considered here, although studying a second-order leader–follower consensus[66] with an underdamped dynamics seems like a natural extension to this work given its acknowledged relevance to collective information transfer in flocks of birds[36,37]. The novelty of these results effectively has implications for a vast breadth of collective decision-making protocols involving a continuum of options to choose from. Firth[18] recently speculated the importance of complex contagions in animal social networks and behavioral contagions. Interestingly, Rosenthal et al.[2] had the intuition that complex contagions occur in schools of golden shiners owing to the high levels of clustering exhibited by their network of interaction. This led them to consider a (fractional) threshold-based model—generating complex contagions—to characterize behavioral cascades in this particular schooling setting[2]. While Rosenthal et al. limited their work to threshold-based models to analyze this phenomenon, our results would allow their study to be expanded upon, as the decision-making process in golden shiners is clearly based on a continuum of options. From the same research group and with the same animal groups, Sosna et al. provided additional empirical evidence about the importance of the topology of the interaction network on collective responsiveness[3]. They also speculate that the fish might actively control their interactions to achieve a higher collective responsiveness[3]. Our results offer a theoretical backing to this idea, and it might provide biologists with new directions to explore and experiment.

Besides shedding a new light on our understanding of collective behaviors, these results have also clear implications for the design of man-made networked systems: e.g., Internet of Things, multi-robot systems, dynamic sensor networks. Finally, we hope the present analysis of these rich phenomena of transition from simple to complex contagion will be extended to more complex networks, including heterogeneous, temporal, and/or multilayer networks.

## Methods

**Linear threshold model**. The LTM is a binary-option model of collective decision-making widely used in the study of complex contagion[13,16]. As described in refs. [16,17,27], it consists of a connected network of $N$ nodes, with each node $i$ characterized by a state variable with two possible discrete values $s_i(t) \in \{0, 1\}$—inactive or active—and a fixed threshold value $\theta_i \in [0, 1]$. At $t = 0$, all nodes start in the inactive state except for a random seed node $l$ and its neighbors, denoted by $\{j \mid j \sim l\}$, whose state is set to active. Whenever the fraction of an agent's neighbors that are in the active state is larger than or equal to the agent's threshold $\theta_i$, that node will switch to the active state, i.e., $s_i(t + 1) = 1$ when $\langle s_{j \sim i}(t) \rangle \geq \theta_i$. The LTM dynamics on a given network can be characterized by the average polarization or activity, defined as $P(t) = \langle s_i(t) \rangle_{\{i=1,\dots,N\}}$, and the associated polarization speed[11,46]

$$v(t) = \frac{P(t) - P(0)}{t}, \tag{3}$$

which effectively is the average propagation speed until instant $t$.

Note that when $P(t) = 1$, all nodes are in the active state, and a global cascade has taken place. Particular emphasis has been put on characterizing which network topologies lead to global cascades for certain values of $\theta_i$[13,16,17], and the realization that complex contagions spread faster on networks with high levels of clustering[16,20,45]. However, numerous empirical observations attest that behavioral cascades are rarely global: evasive maneuvers in schooling golden shiners involve 10–30% of the individuals[2] and the adoption of new health behavior in an online social network experiment only reached 30–60% of participants[20]. Since complex contagions exhibit faster spreading in their early stage of propagation[20,45] that in their late near-global stage, we focus here on the early dynamics of the cascade and therefore consider the polarization speed $v(t_f)$ for which 30% of the nodes are activated, i.e., for the smallest time $t_f$ where $P(t_f) > 0.3$ (Supplementary Fig. S1 for other cascade sizes).

To study the transition from simple to complex contagion with the LTM, we consider fairly large networks of $N = 10^4$ nodes, with identical thresholds $\theta_i = \theta \, \forall \, i$, in order to ensure that the active state propagates for long enough at low values of $\theta$. We let the active state propagate from a randomly selected seed node and its neighbors until the system reaches an equilibrium where $P(t_{\text{end}} + 1) = P(t_{\text{end}})$. In our analysis, we only consider networks where $P(t_{\text{end}})$ is at least as large as the desired cascade size, and we omit from our analysis the simulations where the active state failed to propagate.

**Leader–follower consensus model.** The leader–follower consensus model (LFC)[11,12,42]—a particular version of the Taylor model[30]—is commonly used by the control community to study opinion dynamics and formation control. Here, $N$ identical agents perform a distributed consensus protocol on their state variable $x_i(t)$, and one particular agent $i = 0$—the leader—follows an arbitrary trajectory $x_0(t) = u(t)$ instead. The dynamics of this linear system is governed by the following set of first-order ODEs:

$$\frac{dx_i}{dt} = \sum_{j=1}^{N} w_{ij} x_j(t) + w_{i0} u(t), \tag{4}$$

where $w_{ij} = \omega_0(a_{ij}/k_i - \delta_{ij})$ is the inter-agent consensus protocol weight for the interaction between agents $i$ and $j$. The natural response frequency $\omega_0$ is assumed to be the same for all agents. The degree of agent $i$ is given by $k_i = \sum_{j=0}^{N} a_{ij}$, and the adjacency matrix entry is $a_{ij} = 1$ if agent $i$ is connected to $j$ and 0 otherwise—classically $\delta_{ij}$ is the Kronecker delta. The solution to Eq. (4) in the frequency domain can be expressed in matrix form as[12]

$$\mathbf{X}(\omega) = (j\omega \mathbf{I} - \mathbf{W}_F)^{-1} \mathbf{W}_L u(\omega), \tag{5}$$

with $\mathbf{I}$ the appropriately sized identity matrix, $\mathbf{W}_F = (w_{ij})_{N \times N} = \omega_0 \mathbf{L}_G$ where $\mathbf{L}_G$ is the grounded Laplacian matrix[42] and $\mathbf{W}_L = (w_{i0})_{\{i=1,\dots,N\}}$ is the consensus protocol between the $N$ followers and the leader $i = 0$. The frequency response of the multiagent system measures the ability of the agents to follow the leader's trajectory, $u(t)$, and can be expressed in the frequency domain as the transfer function along the $j\omega$ axis in the $s$-plane[12,67]

$$\mathbf{H}(\omega) = \left(\frac{\delta \mathbf{X}}{\delta u}\right)(\omega) = (j\omega \mathbf{I} - \mathbf{W}_F)^{-1} \mathbf{W}_L, \tag{6}$$

with the entries of the vector $\mathbf{H} = (h_i)_{\{i=1,\dots,N\}}$ corresponding to the individual agent's frequency response. It is clear from Eq. (6) that the response function has a nontrivial dependency on the topology of the agents' connectivity through the adjacency matrix, and therefore the entries of $\mathbf{W}_F$ and $\mathbf{W}_L$. We define the collective frequency response of the system as[12]

$$H^2(\omega) = \mathbf{H}^\dagger(\omega) \mathbf{H}(\omega) = \sigma^2(\omega), \tag{7}$$

with † being the adjoint matrix operator and $\sigma$ being the singular value of the system with a single leader—classically used in the analysis of multiple-input and multiple-output (MIMO) systems. When all agents in the system perfectly follow the leader's behavior the response reaches its maximum value, $H^2 = N$. Thus the quantity $H^2/N$ can be interpreted as the fraction of agents in the system following the leader. It is worth stressing that since $H^2$ is not constant under leader selection, the results we present in this paper are obtained by averaging over all possible leader placements in the network.

To study the transition from simple to complex contagion in the LFC, we consider networks of $N + 1 = 240$ agents. This is significantly smaller than for the LTM, but the LFC shows a much higher sensitivity to variations in the network metrics, which yields a range of $H^2$ large enough to perform our analysis, even with relatively small network sizes.

**Networks and metrics.** To study the influence of the network topology on the behavioral propagation in the LTM and the LFC, we use the small-world Watts–Strogatz (WS) network model[5]. These networks enable us to vary key network metrics by changing a single parameter, namely the rewiring probability[5] $p$. To quantify the generated network topologies, we use the following metrics: the clustering coefficient $C$, the average shortest path $\ell$ and the Kirchhoff index $R_g$ defined below. We use an average degree $\langle k \rangle = 16$ for Figs. 1 and 2a, b to ensure high levels of $C$ before rewiring[60] and a large range in $\ell$ and $R_g$.

*Clustering coefficient.* To measure the level of community structure, and thus the potential for reinforcement of the behavior that is propagating, we use the average of the local clustering coefficient as defined in ref. [5], and we simply refer to it as "clustering coefficient" or $C$ throughout the paper.

*Average shortest path.* The shortest path between a given pair of nodes is the minimum number of hops needed to connect the two. The average shortest path is the average between all pairs of nodes. It is well known that for WS networks, decreasing the distance between nodes both increases the effectiveness of simple contagion[5,16] and the time it takes to reach consensus[14].

*Kirchhoff index.* The Kirchhoff index $R_g$—also known as the resistance distance—is a distance metric obtained by replacing every connection with a $1\,\Omega$ resistor and averaging the resistance between all node pairs[53], thus considering all paths on the network. Further, $R_g$ is directly related to the eigenvalues $0 = \lambda_1 < \lambda_2 < \cdots < \lambda_N$ of the Laplacian matrix[53] and can be expressed as

$$R_g = N \sum_{i=2}^{N} \frac{1}{\lambda_i}. \tag{8}$$

The cohesion of the follower states—under a white noise disturbance—can be measured by the $H_\infty$-norm of the dynamics of Eq. (4)—which is $H^2(\omega = 0)$—and it has been proven that maximizing the cohesion is done by minimizing the Kirchhoff index $R_g$[42,43].

**Correlations.** We consider Spearman's rank correlation $r_s$ to investigate the impact of network metrics on a given collective behavior. We use the rank correlation due to the expected nonlinear relationship between the performance of the collective behavior and network metrics for both the LTM and LFC[12,14] (Supplementary Figs. S2, S5, and S6 for point distributions). The Spearman's correlation is defined by

$$r_s = \frac{\text{cov}(\text{rg}_X, \text{rg}_Y)}{\text{std}(\text{rg}_X)\text{std}(\text{rg}_Y)}, \tag{9}$$

with $\text{rg}_X$ being the rank of the metric, $\text{cov}(X)$ and $\text{std}(X)$ the covariance and standard deviation, respectively.

Given the fact that the WS networks are generated by varying one single free parameter (the rewiring probability $p$), the associated network metrics are inevitably correlated. As a consequence, we are unable to distinguish the individual effects of each network metric for a given average degree $\langle k \rangle$ (insert of Fig. 2b). To overcome this, we generate networks of different average degrees $k \in \{4, 6, \dots, 32\}$, and consider a sampled set of these networks for which $C$ and $R_g$ vary independently of each other (insert of Fig. 2c). However, as shown in refs. [11,12], the collective frequency response $H^2$ is notably influenced by the degree distribution, and so are the network metrics considered $\chi \in \{C, \ell, R_g\}$[14,53,60]. To account for that degree dependency, we normalize all quantities involved, with the normalized quantity denoted by an overbar. First, the normalized collective frequency response is given by

$$\bar{H}^2(\omega, k, p) = \frac{H^2(\omega, k, p)}{H^2(\omega, k, p_{\max})}, \tag{10}$$

in which $p_{\max}(\omega, k)$ is the rewiring probability that yields the largest $H^2(\omega, k, p)$. Note that $\bar{H}^2(\omega) \leq 1$. Second, the normalized clustering coefficient is defined as

$$\bar{C}(k, p) = \frac{C(k, p)}{C(k, p = 0)}, \tag{11}$$

such that for any $k$ the most clustered networks, obtained for $p = 0$, have $\bar{C} = 1$ (Supplementary Fig. S5). Third, since the distance metrics also change with $\langle k \rangle$[14,53], we introduce the following normalized distance metrics

$$\bar{\ell}(k, p) = \frac{\ell(k, p)}{\ell(k, p = 1)}, \tag{12}$$

$$\bar{R}_g(k, p) = \frac{R_g(k, p)}{R_g(k, p = 1)}, \tag{13}$$

which are based on the smallest distances, obtained with $p = 1$.

From all the networks generated with different $p$ and $\langle k \rangle$, we construct a sample that almost fully decorrelates the clustering coefficient and the Kirchhoff index: i.e., such that $r_s(\bar{C}, \bar{R}_g) \approx 0$ (insert of Fig. 2c, and Supplementary Fig. S3 for other subsamples of networks).

**Nonlinear heading consensus experiments.** To empirically measure the collective response—with differently paced leaders—we use the same experimental multirobot setup as ref. [12] with $N = 10$ robotic units (called "eBots") communicating by means of their so-called "swarm enabling unit"[64]. These ground differential-drive robots are collectively moving about a two-dimensional domain (Supplementary Fig. S10). The motion of the units is the superposition of a translational motion and a rotational one. In this setup, the leader agent undergoes a constant rotational motion such that its heading $\alpha_L(t)$ is governed by $\frac{d\alpha_L(t)}{dt} = \omega$,

with $\omega$ denoting the frequency of the leader. Its "leader" status comes from the fact that its behavior does not depend on the follower agents. Note that the agents have no way of distinguishing the leader from any other agent in the system.

The robot's heading is the only state variable driven by the robot's controller regardless of the speed. Specifically, the $N_F = 9$ follower eBots seek to align their heading $\alpha_i(t)$ with that of their neighbors $\alpha_{j \sim i}(t)$. The nonlinear heading consensus algorithm determines a target heading $\overline{\alpha}$ for each unit according to

$$\overline{\alpha}_i = \langle \alpha \rangle_{j \sim i} = \arctan\left(\frac{\sum_{j \sim i} \sin \alpha_j}{\sum_{j \sim i} \cos \alpha_j}\right), \tag{14}$$

where $j \sim i$ denotes the set of topological neighbors of $i$ in the network sense, and $\langle \cdot \rangle$ is an angular average. The nonlinear heading consensus (14) is identical to the one considered in Vicsek's model[63] except for the fact that we use a static topological neighborhood while a metric one was originally considered in ref. [63]. Each follower unit updates its target heading $\overline{\alpha}_i$ asynchronously every $\Delta T = 0.1$ s. The nonlinear heading consensus algorithm used in the experiments is a discrete-time equivalent of

$$\frac{d\alpha_i}{dt} = \omega_0(\langle \alpha \rangle_{j \sim i} - \alpha_i), \tag{15}$$

with $\omega_0 \Delta T \gg 1$ and for times $t \gg \Delta T$. Here, $\omega_0$ is the natural frequency of angular rotation of the eBots. The information of each neighbor's state is also updated with the same sampling rate, but not necessarily concurrently with each other or with the update of Eq. (14).

The robots are interconnected according to three fixed network topologies with $\langle k \rangle = 4$ due to size limitations of the network $N = 10$, namely: the connected caveman network[59] (for maximal clustering coefficient), a $k$-regular random network (for low values of $C$), and the 1D ring lattice (WS networks with $p = 0$ offering an intermediate level of $C$). These three fixed network topologies provide a wide range of values for the clustering coefficient as shown in Fig. 3.

Each run of the experiment starts at $t = 0$ with all eBots aligned with the leader: i.e., $\alpha_i(0) = \alpha_L(0) = 0 \; \forall i$. Each run lasts for a duration of $T = 10$ min to make sure the leader performs a meaningful number of rotations for all frequencies considered. The capacity of the $N_F$ followers to maintain their heading aligned with that of the leader is measured by the empirical collective response[12]

$$\mathcal{H} = \sum_{i=1}^{N_F} \frac{1}{T} \int_0^T \cos(\alpha_i(t) - \alpha_L(t)) dt, \tag{16}$$

which is averaged over three repeated runs, for each angular speed of the leader considered.

## Data availability

All data needed to evaluate the conclusions in the paper are present in the paper and/or the Supplementary Information. Additional data related to this paper may be requested from the authors.

## Code availability

All codes developed for this study are available at https://github.com/Horsevad/Simple_to_complex_contagion_in_collective_decision-making.

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

## Acknowledgements

This work is partially supported by the National Research Foundation (NRF) under its National Cybersecurity R&D Programme (Award No. NRF2014NCR-NCR001-040). The authors also acknowledge the support from the SUTD-MIT International Design Center under the Grant #IDG31900101. A.B. is partially supported by the Agence Nationale de la Recherche (ANR) project DATAREDUX (ANR-19-CE46-0008).

## Author contributions

N.H., D.M., and R.B. designed the study and the experiment. N.H. and D.M. developed the analytical and numerical tools. D.M. and R.B. conceived the experiments. N.H. conducted the numerical calculations and simulations. A.B., R.E.K., D.M., R.B., and N.H. analyzed the data and results. N.H. conducted the experiments. All authors wrote and reviewed the manuscript.

## Competing interests

The authors declare no competing interests.
