## [Peer Review File · Nature Communications]

REVIEWER COMMENTS

Reviewer #1 (Remarks to the Author):

Dear editor and authors,

Many thanks for the opportunity to review this fantastic paper. I enjoyed reading this. This paper extended the traditional threshold-based and binary options based model to a general class of collective decision-making processes. Verifications include simulations and experiments on robotics. Results are quite solid and consistent except for the part of the robot experiments. And I do have several other concerns mainly regarding the interpretations.

Major comments:

1. The novelty is not high to me. Two main contributions are mentioned in this paper. One is the extension of the traditional threshold model. The authors argue that traditional models are most binary options, while they considered continuous options here. However, this idea is not new, for example, the studies from Naomi Leonard, cited as reference No. 40 already considered this. The second contribution is that they find the network topology is critical. This is also not new, as there are plenty of studies on this, such as Sosna et al. 2019 PNAS. and Santos et al. 2005 PRL. The contribution of the understanding of collective behaviour is not clear, as this can only give one hypothesis to describe the real system. I am afraid the authors did an overselling here. This study can not really help us understand the real mechanism behind collective behaviour. One can not give an explanation of collective animal behaviour unless they are working on the biology data. Therefore, I would like to say, this work provides new directions for biologies to explore, test, and verify.

2. The importance of introducing a linear threshold model (LTM) in the first part is not clear. It is also based on two options as the traditional one (see Page 11 Line 350). They claim the speed of contagion is a new variable to tell the simple or complex contagion. However, I think this might not be solid, as the variations of speeds of contagion might come from the fact of simple or complex contagion. I think this is a necessary but not sufficient condition. Namely, is it true that any network with a lower speed of contagion is a simple contagion? And a high speed of contagion is a complex one?

3. The importance/necessity of the experiments on robotics is also not clear. Limited by the physical system, the parameter range is not fully explored. And most discussions are actually based on complementary simulations. I am excited to see the robotic work, but it looks like the robot system is not ideal for this question due to the physical limitation.

4. The introduction of the robot experiments is way too simple. The experimental setup and process are not clear. For example, how the Vicsek Model inspired the angular heading protocol in Eq(12)? In Vicsek Model, geometry distance neighbour selection is applied, how is this situation? How does the ω affect the follower? Are the robots moving in 2D space or just moving locally as a spin? If the robot moves, how the speed of the robot is controlled? Why there is no error bar for the Caveman network while there are error bars for the other networks. Is there any noise for the simulations in Fig 3(b), if there are what are the noises and why there are no error bars either?

Minor comments:

1. In the abstract. The authors claimed the new model has no relay on the threshold. But the LTM in the paper is a threshold-based model, no?

2. Page2 Line40: it is not clear by saying "another general class"

3. Page2 Line52: the last sentence needs citations

4. Page3 Line79: clarify "effectiveness of the behavioural propagation", what is the definition in this paper?

5. Figure 1: the symbol of polarization speed is different from the one in the main text.

6. Page6 Line197: "not shown here", then where to find?

7. Page8 Line251: "...this experimental setup and methodology is identical to the one reported in Ref 11, except for the fact that different networks topologies are considered here." First, It should be "this experimental setup and methodology are identical...". Second, I would suggest giving a brief description of the setups here, because 1) this will save time for those readers who did not read your previous publications. 2) the differences also include the number of robots. In Ref 11 there are 11 robots while there are 10 here.

Reviewer #2 (Remarks to the Author):

In this paper the authors show how the phenomenology of contagion, the transition from simple to complex, is not limited to threshold-based models but it "can also be observed and studied in other different models which are devoid of any activation thresholds or non-linearity". They developed an approach driven by some numerical and theoretical results on small-world Watts-Strogatz networks which then they export to the special case of a little network of robot swarms.

In order to prove their conclusions, first of all the authors consider the transition from simple to complex contagion observed in the linear threshold models as the activation parameter increases. They look at the Spearman's rank correlation (able to reveal a monotonic dependence in the nonlinear case) between some typical topological features of the networks (clustering coefficient, average short path, Kirchhoff index) and the polarization speed at which a random activator node and its neighbours can activate a 30% of nodes.

They observe an inverted trend switching from simple to complex contagion: in simple contagion they find a negative correlation coefficient between the network metrics and the polarization speed, on the contrary the correlation is positive in the complex case.

This suggested to the authors a different way of characterizing the nature of the contagion that can be generalized to different models such as the leader-follower consensus model.

In this model the state of a node is described by a continuous variable so, in absence of a precise definition of "active node", the authors look at the collective frequency response of the system (easily obtained thanks to the linearity of the model) which can play the same role of polarization speed used for the analysis of the transition in the threshold-based models. Therefore, based on previous results, the authors exploit this analogy in order to distinguish simple from complex contagion by analyzing the dependence on the frequency of the Spearman's rank correlations between the response and the network metrics.

But, since the Watts-Strogatz networks have only one free parameter, the network metrics are strongly correlated. Therefore, a "smart" networks sampling with different average degree $\langle k \rangle$ and a proper normalization of the observables considered are necessary to homogeneous comparison and to "dismantle" the correlation between the clustering coefficient and the Kirchhoff index in order to distinguish the role of the single metric in determining the nature of the contagion. In this way the authors obtain that, at low frequencies, the response is highly correlated with Kirchhoff index and few correlated with the clustering coefficient, while the opposite occurs at high frequencies: these results are fully compatible with those relating to threshold-based models.

Finally, they look at some little networks of rotating robots which obey the Vicsek-like rule.

They find complex contagion in an intermediate frequency range and a most effective response with the most clustered networks.

In my opinion, the work is rigorous enough, the manuscript is correctly written and the conditions and conclusions are clear and straightforward. The methodology sounds good and meets the scientific standard. I appreciate the supplementary information too which actually shows well how robust the approach developed by the authors is.

So, I have only few remarks of general nature:

1) I feel that the paper is too long and I'm not sure that the robot swarm experiments are really necessary: from my point of view the main noteworthy result significant to the field is the possibility of generalizing the concept of simple and complex contagion through the analysis of the response function and its correlations with network metrics: can the authors cut something or propose more compelling arguments to justify more than two pages in the main body devoted to robot swarm experiments?

I would focus on the results about scale-free networks which we know are much more similar to real networks. The authors spend for this only one sentence in the main section and one figure in SI without any details and then it is hard to reproduce the results: probably it is appropriate to add something.

2) I'm far from mastering the subject but, basically, it is quite easy to understand this work once you have precise definitions of the quantities and models considered.

About this, I found it frustrating to be forced to look at the references and at the methods section in order to get a precise idea of the work done by the authors.

I believe that something can be put in the main body such as the update rule for LTM and LFC and the definitions of polarization speed and collective frequency response. Sure, the authors can keep the definitions of the network metrics in the method section but I would spend some words to remember the known results on WS networks.

3) What do authors expect from an "underdamped" dynamics governed by a second order ODE of LFC? I mean the simple model of harmonic oscillator obtained by substituting the second derivative of x_i to the left side of eq. 2) and, eventually, by adding a damping term too. In a way (continuous limit with local interaction) we are moving from the heat equation (with damped and diffusive information propagation) to the wave equation (with linear and no damped information propagation) and therefore the two models have clearly different behaviors. Can we expect something new and/or different in the simple/complex contagion scenario?

I believe that a few words on the subject can help to make the results even more general and interesting. Moreover, the collective frequency response function is obtained analytically in a simple way in this case also, so it doesn't seem complicated to have some numerical results by exploiting the network metrics already computed by the authors on the LFC model. However, it is a request without "penalty", I only ask the authors to think about it, to give me their opinion and, eventually, to add something in the paper if they are reasonably convinced to improve it.

Finally, in my opinion, minor changes only are necessary in order to deserve the publication on Nature Communications.

Reviewer #3 (Remarks to the Author):

The manuscript argues that complex contagion is a general feature of models of consensus spreading, not limited to threshold-based models with binary options. Instead, a transition or crossover from simple to complex contagion is observed also in models with continuous state variables, although it is strongly influenced by network topology and the intrinsic

frequency of the spreading behaviour.

To reach this conclusion the authors study three cases. The first is the linear threshold model (LTM) on the Watts-Strogatz small-world network. This is a model with binary options where a site can flip from state 0 to state 1 if a fraction of its neighbours larger than a threshold θ is in state 1. The network is initialised in state 0 except for one site and its neighbours which are set to 1, and the spread of state 1 across the network is monitored. The responsiveness of the system is quantified through the polarisation speed (which measures how long it takes for a given fraction of the network to switch to state 1), and this is measured for networks with average degree $k=16$ and various rewiring probabilities so that they have different clustering coefficient (C), average shortest path (l) and Kirchoff index (R_g). It is observed that for low thresholds the speed increases with decreasing l , C and R_g , while at large threshold this trend no longer holds (indeed, it is reversed for some intervals of l , C , and R_g). This is quantified using Spearman's rank correlation coefficient, which changes sign from negative at low thresholds to positive at large thresholds. This inversion is interpreted as a transition from simple to complex contagion (Fig 1).

The second case is the leader-following consensus (LFC) model, where the individual site's state is given by a real number $x_i(t)$. A leader executes an arbitrary trajectory $u(t)$, and the rest of the nodes follow it with a dynamics given by a first-order ordinary differential equation (eq 2), which reduces to a forced overdamped harmonic oscillator for the case of a single follower. The model is studied again on the small-world network. Here the metric for the network responsiveness is the linear response $H^2(\omega)$. A simple-to-complex transition is found where the Spearman rank correlation shows that at low frequencies response is higher for lower l and R_g (with no correlation to C), while at high frequencies response is higher for higher l and C (and low, but still negative, correlation with R_g). Here networks of different average degree are studied so that the three metrics (l , C , R_g) are not tied to each other.

The third model (which is both simulated and actually implemented in hardware) is a swarm of ten robots which can tune their orientation heading so that it aligns with their neighbours. One robot is chosen as leader and is made to turn its heading with constant frequency, while the rest follow their neighbours with a nonlinear law (eq 12). Three different interaction networks are studied. The response is quantified with a different response function (eq 13). The response is found to decrease with increasing frequency for all networks, but while at low frequencies the response is the same for all networks, there is a range of frequencies where the network with the highest C has a clearly larger response. Higher frequencies cannot be studied experimentally, but simulations show that at high frequencies all networks again have the same response.

In my opinion the manuscript needs significant improvement to make a stronger case for the conclusion, and to be clearer in its message. The main weaknesses I find in the manuscript are:

1. The definition of complex contagion is not clear. In the introduction the authors write that in simple contagion "behavior propagates through a single exposure or interaction", while complex contagion needs "social reinforcement", which is explained as transmission that "requires an individual to have contact with two or more sources of activation". One would expect a more quantitative definition of complex contagion, at least one that is applicable to the models considered here. Or at least, a discussion of how this qualitative definition translates to the metrics that will be later employed to detect the nature of contagion. It is surprising to read in the discussion (p9 l304) that "one is lacking a definite way of characterizing the behavioral contagion". This seems to cast doubt on the

conclusions the authors are drawing! This phrase suggests that there are conceptual problems related to the very definition of complex contagion. This is fine: the present results may help resolve these problems. But given that complex contagion is the central element of the manuscript, these issues should be dealt with up-front and made clear from the beginning.

2. The authors argue that a transition from simple to complex contagion is probably a general feature of a broad class of collective decision-making processes. But the simple-to-complex contagion transition is studied in three models using a different metric for each model. The case would be stronger if the same metric were shown for all models. For example, it seems that the polarisation speed concept can be generalised to the LFC: One starts with all sites at $x=0$, fixes the leader at $u=10$ (say) and counts the number of sites that have exceeded some threshold value. Or a linear response similar to that employed for the LFC could be used for the robots. If this is not possible due to technical reasons, these reasons should be made explicit, and some argument given as to why the metrics chosen can be regarded as equivalent.

3. The swarm experiment is presented as a third scenario where complex contagion is found. Although it seems that this case is quite different from the first two (LTM and LFC models), the robots interact on a fixed network, so that this "swarm" is actually a sort of LFC but with a nonlinear contagion law (eq 12 vs eq 2). A key characteristic of swarms is that as individuals move, the interaction network changes (since the interactions are local); this crucial ingredient is one of the reasons the study of such systems is so challenging. In the present case, the positions of the robots play no role: neighbouring robots (in the network sense) interact only through their orientation (heading), eq 12. In an actual swarm, orientation variations lead to variations of position that in turn can cause changes in the interaction network. This coupling between positions, orientations and interaction network is absent in the model studied, and in this sense the term "swarm" is misleading. "Robot network" would be a more appropriate term.

4. The robot results do not seem to show simple-to-complex transition, at least in the way this transition was shown in the first two models. Instead of an inversion of the trends with C , l , R_g , the response is either independent of the network or has a definite trend. Also, it is not possible to tell which of the network metrics is the main factor influencing the response.

Other issues/comments:

1. The simple-to-complex transition is studied in networks of fixed size (10000 nodes for the LTM, 240 nodes for the LFC). Networks of different sizes would be needed to have a hint on whether e.g. the right panel of fig 1 would show a sharper or less sharp transition for larger networks; similarly for fig 2. Also, fig S1 shows that the threshold value at which the transition happens is significantly affected by the choice of cascade size: $\theta=0.2$ is complex at 10% cascade size, but simple beyond 50%. Here also studying different network sizes would help clarify whether the situation tends to be sharper for larger networks.

2. In fig 1, systems that do not reach 30% contagion are excluded. How does that affect the results? In fig S1, the effect of changing the 30% threshold to other values is studied, but it is not clear why the red curve always stops at $\theta=0.3$. In the caption of fig 1 it is stated that some lines stop prematurely because the 30% cascade size is never reached; one would then expect that changing the cascade size would also change the point at which the lines must stop.

3. The units of $\nu(t)$ (polarisation speed) are, according to eq 1, the inverse of the time unit, rather than length divided by time. The name "polarisation rate" is thus perhaps more appropriate for this quantity.

4. It is stated that the only parameter of WS networks is the rewiring parameter p . In fact, the average connectivity is also a parameter that controls network topology, which is actually also changed by the authors when studying the LFC model. The authors explain that it is necessary to study networks of different degrees to untangle the effect on contagion of the different network metrics C, I, R_g . The question is why this is done only for the LFC and not for the LTM.

5. The expression "pace of the intrinsic behavior" used in the abstract is not very clear to me. Do the authors mean frequency of the leader movement?

Transition from simple to complex contagion in collective decision-making

NCOMMS-21-23105-T

Nikolaj Horsevad, David Mateo, Robert E. Kooij, Alain Barrat, Roland Bouffanais

Note: all edits made to the original version of the manuscript are highlighted in blue for additions and in red for deleted items in the text appended at the end of this document.

Comments by Reviewer #1

Reviewer #1 (Remarks to the Author):

Dear editor and authors,

Many thanks for the opportunity to review this fantastic paper. I enjoyed reading this. This paper extended the traditional threshold-based and binary options based model to a general class of collective decision-making processes. Verifications include simulations and experiments on robotics. Results are quite solid and consistent except for the part of the robot experiments. And I do have several other concerns mainly regarding the interpretations.

We would like to thank Reviewer #1 for the positive comments, and for qualifying our paper as being “fantastic”. We also appreciate that Reviewer #1 assesses our theoretical/simulation results as solid. We have taken good note that our analysis and interpretation of the experimental robotic results should be improved. Following this recommendation we have revised that part of our manuscript, stressing how these robotic experiments supplement and strengthen the theoretical results (see details below).

Major comments:

1. The novelty is not high to me. Two main contributions are mentioned in this paper. One is the extension of the traditional threshold model. The authors argue that traditional models are most binary options, while they considered continuous options here. However, this idea is not new, for example, the studies from Naomi Leonard, cited as reference No. 40 already considered this.

We would like to stress that although our study starts with the LTM, it is not meant to be an extension of this model in any way. Our primary focus is to gain insight into how the spread of behavior affects consensus-based collective decision-making in groups. Some recent works by biologists (e.g., by Couzin and collaborators, Firth and collaborators) have hinted at the possibility of complex contagion occurring in groups driven by continuous consensus—schools of fish and flocks of birds. However, we know remarkably little about the theoretical aspects underpinning such collective information processing in networked systems. Our work addresses this gap and offers some additional theoretical and experimental support to the work of biologists gathering empirical evidence about collective decision-making in animal groups.

Since the phenomenology of transition from simple to complex contagion has so far been limited to the LTM, it appeared logical for it to be the starting point of our study centered on this very transition based on consensus models. To further clarify, our work does not aim at extending any model but instead uses two archetypal models of collective decision-making—the LTM and the LFC—that have both been widely studied. As such, we can benefit from their mathematical simplicity along with a vast breadth of results available in the literature.

We agree with Reviewer #1 about the importance of the work by Naomi Leonard and collaborators (now Ref. [44]) about the extension to the LTM. This was only briefly cited in our original manuscript and we have now added the following statement:

“It is worth highlighting that the original LTM with binary options has been extended to a continuous threshold model (CTM) of cascade dynamics, which involves nonlinearities and a threshold [44]. However, no evidence of a transition from simple to complex contagion has been reported for the CTM.” As

mentioned above, our work does not aim at extending the LTM but instead at extending the concept of *transition* from simple to complex contagion, which has been limited to the LTM so far.

The second contribution is that they find the network topology is critical. This is also not new, as there are plenty of studies on this, such as Sosna et al. 2019 PNAS. and Santos et al. 2005 PRL.

We also agree with Reviewer #1 about the critical role played by the network topology on cascade dynamics. We concur that this is not something new, and indeed many papers have been published on this key topic. We acknowledged that fact in our original submission (Introduction): “Unsurprisingly, the network topology plays a pivotal role in this study.” and following sentences.

One could even argue that the importance of the network topology was one of the key contributions by Centola & Macy in their seminal paper (Ref. [13]) from 2007, where the concept of complex contagion with the LTM has been introduced. Some of our previous works also added to this important topic of the influence of network topology on collective decision-making (Ref. [11-12]). However, our main contribution here is about the influence of the network topology on the *transition* from simple to complex contagion with a collective decision-making process devoid of binary options and threshold. To improve the way we are conveying the main contributions of our study, we have significantly edited the Introduction (in particular the two paragraphs related to these contributions, see manuscript with highlighted changes appended at the end of this response letter).

We also added Santos et al. (2005) PRL. (Ref. [16]) to our references. As for Sosna et al. 2019 PNAS. (Ref. [3]), we have added some dedicated comments at several locations given how relevant this reference is. We would like to take this opportunity to thank Reviewer #1 for highlighting it to us. Some of the added statements are:

“For instance, Sosna et al. recently reported a study on the “fear response” of a school of fish collectively making fast decisions under risky conditions [3]. They found that the properties of the network (their “social connectivity”) are the primary factors responsible for the high collective responsiveness of the school in terms of number of behavioral cascades and their sizes.”

and

“Moreover, as with all ethological results, it is possible to modify the interaction among agents in some ways [3], yet it is virtually impossible to fully control all aspects of it.”

and

“From the same research group and with the same animal groups, Sosna et al. provided additional empirical evidence about the importance of the topology of the interaction network on collective responsiveness [3]. They also speculate that the fish might actively control their interactions to achieve a higher collective responsiveness [3]. Our results offer a theoretical backing to this idea, and it might provide biologists with new directions to explore and experiment. ”

Finally, to close this point, we would like to stress that neither Santos et al. (2005) PRL. (nor Sosna et al. (2019) PNAS (Ref. [15] and [3])) have dealt with the influence of the network topology on the *transition* from simple to complex contagion.

The contribution of the understanding of collective behaviour is not clear, as this can only give one hypothesis to describe the real system. I am afraid the authors did an overselling here. This study can not really help us understand the real mechanism behind collective behaviour. One can not give an explanation of collective animal behaviour unless they are working on the biology data. Therefore, I would like to say, this work provides new directions for biologies to explore, test, and verify.

We recognize that some statements about the direct implications of our results on the dynamics of animal and social groups should be more carefully conveyed. Given the mathematical foundations of our work and the robotic experiments, one cannot indeed make any direct claims about biological or social systems. Instead, we merely intended to state that our results could serve biologists and social scientists to: (1) reconsider previous studies where contagions occurred in consensus-based collective decision-making and check if our results could shed a new light on the published results (e.g., Rosenthal et al. (2015) PNAS (Ref. [2])), and (2) identify new research questions and design new studies to obtain empirical data on these collective behaviors. Clearly, such experiments and the associated empirical data are necessary to formulate any claim for biological/social systems. To address this point, we have reformulated several statements. Specifically, we have modified:

- the last sentence of the Abstract.
- the last sentence of Paragraph 5 of the Introduction. (~ lines 59-61 in main, 59-64 in diff).
- Paragraph 9 of the introduction. (~ lines 98-109 in main, 103-118 in diff).
- First paragraph of Section titled “Complex contagion with networked robots governed by nonlinear heading consensus”. (~ line 233 in main, 245 in diff).
- Third paragraph of Section titled “Complex contagion with networked robots governed by nonlinear heading consensus”. (~ lines 256-258 in main, 267-269 in diff).
- Penultimate paragraph of the Discussion section. (~ lines 379-381 in main, 395-399 in diff).

2. The importance of introducing a linear threshold model (LTM) in the first part is not clear. It is also based on two options as the traditional one (see Page 11 Line 350).

We appreciate the feedback about the lack of clarity in our motivation to start with the linear threshold model (LTM). As stressed in our Abstract and Introduction, our study focuses on the rich phenomenology of simple and complex contagions, as well as the transition from one type of contagion to the other. The review of the literature on this topic unambiguously reveals that the traditional LTM has been (and still remains) the primary collective decision-making protocol investigated when looking at complex contagion and its transition from a simple one. This started with the seminal work of Centola & Macy in 2007 (Refs. [13,15]), and ever since the LTM has continued to be the central focal point when considering complex contagions (see Guilbeault, D., & Centola, D. (2021) Nat. Comms. Topological measures for identifying and predicting the spread of complex contagions. Nature communications, 12(1), 1-9.). As previously mentioned, our goal in this work is not to extend the LTM, but instead to extend the realm of phenomena involving a transition from simple to complex contagion. Specifically, our emphasis is on another general class of collective decision-making that has relevance to a wealth of collective behaviors in nature and engineered systems, namely the linear consensus protocol. Having said that, we felt it important to bridge the knowledge gap between the vast amount of work dealing with simple/complex contagions based on the LTM, and our extension of this phenomenology to consensus-based protocols lacking thresholds and nonlinearities. Hence, the first set of results in our manuscript (see Fig. 1) are for the LTM.

Furthermore, it is worth adding that even though the LTM with complex contagions has received notable attention over the past 15 years, the transition from simple to complex contagion has received substantially less attention, and is far from being fully understood. Hence, there are still some elements to be analyzed and some effects to be interpreted. For instance, our analysis of the speed of propagation of partial/incomplete cascades and the associated correlations with particular network metrics represent an additional step in that direction.

To clarify this point, we have added the following statement to the Introduction: “This is achieved by characterizing which type of contagion is driven by what network metric with two prototypical models of collective decision-making: a threshold-based one and a consensus-based one.” We have also added a statement to the Results section (“Polarization speed in threshold models”): “It is worth stressing that the investigation of this important statement remains limited to the long-term dynamics of global cascades [45]. On the other hand, the early dynamics of contagion affecting a smaller fraction of nodes—say cascades of 30% activated nodes, which is still macroscopic—has been relatively overlooked. There has been no attempt to relate the actual speed of contagion with the transition from simple to complex contagion for incomplete—i.e., non-global—cascades.”

They claim the speed of contagion is a new variable to tell the simple or complex contagion. However, I think this might not be solid, as the variations of speeds of contagion might come from the fact of simple or complex contagion. I think this is a necessary but not sufficient condition.

This is a very important point. Our argument is in fact not about the actual speed of contagion, whether it is lower or higher, but is about the variations/trend of this speed of contagion when varying the network properties at a **fixed** threshold value. This latter point is extremely important. For instance, let us turn to Fig. 1 and consider $\theta = 0.1$. It is well-known that at such a low threshold value, the contagion

is simple. Our analysis of the speed of contagion v reveals that this quantity increases when the network distance decreases (ℓ or R_g). On the other hand, when considering a higher threshold value, say $\theta = 0.3$, it is also well-known that the contagion is complex. In that case, the speed of contagion increases v with the clustering coefficient C .

Namely, is it true that any network with a lower speed of contagion is a simple contagion? And a high speed of contagion is a complex one?

These two questions help us make our point; this is actually not true. Let us carefully inspect Fig. 1 again, and let us consider a polarization speed $v \simeq 10^{-3}$. For such a low value of v (purple color with inverted triangle symbols in Fig. 1), we have a simple contagion with $\theta < 0.1$ and a complex contagion with $\theta > 0.3$. To further elaborate on our response to the previous point, what matters here is the relative position of the purple curve with respect to the green, blue and red one. At low θ , the purple curve represents the worst performing (simple) contagion. However, when increasing the threshold, it becomes the best performing (complex) contagion, being above the green, blue, red ones (in that order).

We believe that the explanation above would help the reader better understand the analysis in terms of the relationship between speed of consensus and nature of the contagion. We have therefore added the following statement to the Results section:

“These trends can also be appreciated by observing the particular network topology corresponding to the purple curve (high C , ℓ and R_g) in Fig. 1(a). It goes from producing the worst performing simple contagion (out of the 4 topologies considered here) at low θ , to generating the best performing complex contagion at higher threshold values.”

3. The importance/necessity of the experiments on robotics is also not clear. Limited by the physical system, the parameter range is not fully explored. And most discussions are actually based on complementary simulations. I am excited to see the robotic work, but it looks like the robot system is not ideal for this question due to the physical limitation.

We agree with Reviewer #1 that our original manuscript lacked a justification for the need of robotic experiments. The key point here is that to the best of our knowledge, complex contagions have never been observed in networked robotic systems. By tuning the network topology interconnecting the robots performing a nonlinear consensus, we obtain (without the need of our complementary simulations) the confirmation of the existence of such complex contagions. In addition, a recent review paper by Dorigo, Théraulaz & Trianni (2021) Proc. IEEE, (Ref. [61] in the manuscript) stresses the important fact that the key issue of simulation-reality gap in robotics is exacerbated with robot swarms, where many robots have to interact with each other.

First, our multi-robot system aims to achieve a higher modeling fidelity of collective motion involving nonlinear heading consensus. Our theoretical results were obtained for a linear consensus (LFC) thanks to some powerful concepts borrowed from the theory of linear systems. When considering a collective dynamics based on a nonlinear consensus, this theoretical framework no longer applies, and one has to turn to either simulations or physical experiments. We now justify the necessity for us to use physical experiments given the inevitable low fidelity associated with simulations. The following paragraph has been added:

“Although simulations offer unique ways to systematically analyze algorithms of collective dynamics, they inevitably reduce the fidelity of the model to achieve computational tractability. Indeed, the simulation-reality gap in robotics is known to be exacerbated with multi-robot systems [61]. For instance, simulations can fail to adequately capture: (1) the complex physical interactions among agents, (2) the inherent variability among units forming the group, and (3) the fine details of real-world settings in which the agents are embedded. As a matter of fact, achieving high-fidelity simulations often requires the input or feedback from physical experiments.”

We have also changed the heading of that Section to reflect the emphasis on nonlinear consensus with our multi-robot system. Furthermore, we also give additional details about the importance of nonlinear heading consensus based on very recent empirical data obtained from the analysis of traveling groups of pigeons [62]:

“Data gathered *in robotico*, with a highly controlled and controllable environment, enable a rapid investigation of a number of hypotheses about collective behaviors. In turn, the outcome of such investigations

could serve biologists to identify new directions to explore, test and validate with empirical data. Following this strategy, we use a networked robotic system to assess various socially transmittable behaviors when changing the topology of the interaction network in the presence of a collective decision-making with a nonlinear component. It is worth stressing that the results obtained with the LFC (Fig. 2) are for a linear system dynamics, yet they reveal a surprisingly complex phenomenology. Nonetheless, some collective decision-making processes among moving animals are based on a consensus associated with the direction of travel, and are inherently nonlinear. Such nonlinear interactions among conspecifics have recently been shown to be responsible for sudden directional switches in groups of pigeons [62].”

Although simple contagions have been generated and commonly observed in multi-robot systems for a long time, empirical evidence of complex contagions in robotics has been missing. Therefore our primary interest was the observation of the complex contagion, observed in the intermediate frequency range $0.06 \text{ Hz} \leq \omega \leq 0.2 \text{ Hz}$. The confirmation of the existence of a complex contagion in our physical testbed is obtained without the need for simulations. However, to give a complete picture of this phenomenon at a much higher frequency, we felt that simulations could augment the experiments. It is worth stressing that the collective behavior at these high frequencies is of very limited interest compared to the intermediate frequency range. Indeed, at high frequency, the collective frequency response drops rapidly with increasing frequency. This fact has been reported by our group in Mateo et. al., (2019) Sci. Advances (Ref. [12]).

4. The introduction of the robot experiments is way too simple. The experimental setup and process are not clear. For example, how the Vicsek Model inspired the angular heading protocol in Eq(12)? In Vicsek Model, geometry distance neighbour selection is applied, how is this situation? How does the ω affect the follower? Are the robots moving in 2D space or just moving locally as a spin? If the robot moves, how the speed of the robot is controlled? Why there is no error bar for the Caveman network while there are error bars for the other networks. Is there any noise for the simulations in Fig 3(b), if there are what are the noises and why there are no error bars either?

As mentioned in our original submission, the experimental setup is identical to the one reported in our article published in Mateo et. al. (2019) Science Advances (Ref. [12]). The only difference (besides the fact that we use one less robot to accommodate the peculiarities of the caveman network) is the topology of networks considered. Here, we consider three distinct network topologies (Random, Ring & Caveman) offering a wide range of values for the clustering coefficient. However, to make the manuscript more self-contained, we have added additional details about the setup and methodology in the Methods subsection dedicated to the experiments. In particular, we have added one equation (new Eq. (12)) and expanded the discussion about the connection with Vicsek’s model. Reviewer #1 is right that the original Vicsek model uses a metric neighborhood, whereas here we use a topological neighborhood. In other words, any robotic unit performs a nonlinear heading consensus based on the heading of its neighbors in the network sense (for all 3 topologies considered here: ring, random and caveman). This point is now more clearly explained. The dynamics of the leader and how ω controls it is also detailed based on elements borrowed from our Ref. [12]. Details about the 2D movement and the robot’s controller have also been added.

Regarding the error bars for the Caveman network: they are actually present but they are small, and in the absence of whiskers, they are hidden by the green bullet symbol. We have now increased the size of the whiskers and they are now visible. As for the simulations, there is no stochastic element in them and therefore there is no need to perform any statistical analysis.

Minor comments:

1. In the abstract. The authors claimed the new model has no relay on the threshold. But the LTM in the paper is a threshold-based model, no?

As highlighted in our response to point 2. above, the classical LTM (wich is indeed a threshold-based model) has been our starting point to bridge the knowledge gap with the LFC, which is a prototypical model of collective decision-making based on consensus. The edits detailed in our response to point 2. should address this minor comment.

2. Page2 Line40: it is not clear by saying “another general class”

We have rephrased the end of the sentence to clarify this point:
“Here, we show theoretically and experimentally that such a transition from simple to complex contagion, as originally identified in threshold-based models, can also be exhibited by another general class of collective decision-making processes; specifically a class of models based on consensus and devoid of any thresholds or nonlinearities.”

3. Page2 Line52: the last sentence needs citations

We have added three references: one is a review paper by Proskurnikov & Tempo (2017) Annual Reviews in Control (Ref. [30]), which was present in our original version, and the other two are very relevant papers: Friedkin et al. (2016) Science (Ref. [28]), and Tian et. al. (2021) IEEE-TAC (Ref. [29]).

4. Page3 Line79: clarify "effectiveness of the behavioural propagation", what is the definition in this paper?

We have added the following details:
“This transition is made apparent by measuring the effectiveness of the behavioral propagation—quantified here by means of the concept of collective frequency response [12]—when varying the topological features of the interaction networks...”

5. Figure 1: the symbol of polarization speed is different from the one in the main text.

Thanks for pointing out this inconsistency. We have now updated the font on all the relevant figures, such that ‘v’ matches the main text.

6. Page6 Line197: "not shown here", then where to find?

We have added a supplementary Figure, and changed the text to “(see Supplementary Fig. S9)”

7. Page8 Line251: "...this experimental setup and methodology is identical to the one reported in Ref 11, except for the fact that different networks topologies are considered here." First, It should be “this experimental setup and methodology are identical...”. Second, I would suggest giving a brief description of the setups here, because 1) this will save time for those readers who did not read your previous publications. 2) the differences also include the number of robots. In Ref 11 there are 11 robots while there are 10 here.

We have now changed “is” to “are”. Based on this recommendation, we have added additional details about the setup and methodology in the Methods subsection dedicated to the experiments. This indeed makes our manuscript more self-contained.

As for the change in the number of robots from 11 down to 10, it was made on purpose because of the particular nature of the caveman network. Specifically, it is impossible to divide 11 into an equal number of integers as required here. As a consequence, we removed one robot from our system, and we stuck with that number for the other network topologies considered.

Comments by Reviewer #2

Reviewer #2 (Remarks to the Author):

In this paper the authors show how the phenomenology of contagion, the transition from simple to complex, is not limited to threshold-based models but it “can also be observed and studied in other different models which are devoid of any activation thresholds or non-linearity”. They developed an approach driven by some numerical and theoretical results on small-world Watts-Strogatz networks which then they export to the special case of a little network of robot swarms. In order to prove their conclusions, first of all the authors consider the transition

from simple to complex contagion observed in the linear threshold models as the activation parameter increases. They look at the Spearman's rank correlation (able to reveal a monotonic dependence in the nonlinear case) between some typical topological features of the networks (clustering coefficient, average short path, Kirchhoff index) and the polarization speed at which a random activator node and its neighbours can activate a 30% of nodes.

They observe an inverted trend switching from simple to complex contagion: in simple contagion they find a negative correlation coefficient between the network metrics and the polarization speed, on the contrary the correlation is positive in the complex case.

This suggested to the authors a different way of characterizing the nature of the contagion that can be generalized to different models such as the leader-follower consensus model.

In this model the state of a node is described by a continuous variable so, in absence of a precise definition of "active node", the authors look at the collective frequency response of the system (easily obtained thanks to the linearity of the model) which can play the same role of polarization speed used for the analysis of the transition in the threshold-based models. Therefore, based on previous results, the authors exploit this analogy in order to distinguish simple from complex contagion by analyzing the dependence on the frequency of the Spearman's rank correlations between the response and the network metrics.

But, since the Watts-Strogatz networks have only one free parameter, the network metrics are strongly correlated. Therefore, a "smart" networks sampling with different average degree $\langle k \rangle$ and a proper normalization of the observables considered are necessary to homogeneous comparison and to "dismantle" the correlation between the clustering coefficient and the Kirchhoff index in order to distinguish the role of the single metric in determining the nature of the contagion. In this way the authors obtain that, at low frequencies, the response is highly correlated with Kirchhoff index and few correlated with the clustering coefficient, while the opposite occurs at high frequencies: these results are fully compatible with those relating to threshold-based models.

Finally, they look at some little networks of rotating robots which obey the Vicsek-like rule. They find complex contagion in an intermediate frequency range and a most effective response with the most clustered networks.

In my opinion, the work is rigorous enough, the manuscript is correctly written and the conditions and conclusions are clear and straightforward. The methodology sounds good and meets the scientific standard. I appreciate the supplementary information too which actually shows well how robust the approach developed by the authors is.

So, I have only few remarks of general nature:

We would like to thank Reviewer #2 for her/his overall positive assessment, and for finding our work rigorous, clear and straightforward. We appreciate the extensive summary of our work, which reflects a full understanding of our analysis and results.

1) I feel that the paper is too long and I'm not sure that the robot swarm experiments are really necessary: from my point of view the main noteworthy result significant to the field is the possibility of generalizing the concept of simple and complex contagion through the analysis of the response function and its correlations with network metrics: can the authors cut something or propose more compelling arguments to justify more than two pages in the main body devoted to robot swarm experiments?

We agree that the central result of our work is the one about the existence of the transition from simple to complex contagion with the LFC (Fig. 2). Having said that, this theoretical result is obtained for a linear system based on the linear theory (since we use the linear leader-follower consensus). One could

therefore object to the fact that a large number of consensus in real systems (including animal motion, bird flocks, etc.) are based on a nonlinear one. In our opinion, this calls for a verification that complex contagions still occur with a nonlinear consensus protocol. Furthermore, the extension of this key result to a nonlinear heading consensus would offer biologists new directions to explore and test.

As this research question cannot be investigated theoretically (the linear theory no longer applies), we are left with simulations or experiments. Although simulations of multi-agent systems are certainly much easier to implement and use, their limitations are commonly and frequently acknowledged. For instance, in a recent review paper, Dorigo, Théraulaz & Trianni (2021) Proc. IEEE (Ref. [61]) stress the importance to carry out physical experiments given the fact that the simulation-reality gap issue in robotics is exacerbated with multi-robot systems. For all these reasons, we feel that these experiments and the associated results deserve some space in this work and manuscript.

Nonetheless, we have edited our manuscript to make a more compelling case about the need for these robotic experiments. Specifically, we have changed the heading of that Section to reflect the emphasis on nonlinear consensus with our multi-robot system. Furthermore, we also give additional details (see the added second paragraph in that Section) about the importance of nonlinear heading consensus based on very recent empirical data obtained from the analysis of traveling groups of pigeons [61]. We also discuss the importance of robot experiments given the known challenges due to the simulation-reality gap when working with multi-agent systems.

I would focus on the results about scale-free networks which we know are much more similar to real networks. The authors spend for this only one sentence in the main section and one figure in SI without any details and then it is hard to reproduce the results: probably it is appropriate to add something.

We appreciate the perspective of the Reviewer about the importance of the results with scale-free networks. We felt the same way and this is why we included those scale-free results in the SI. As the Reviewer can easily imagine, there are a large number of effects that can be studied when moving from homogeneous networks (here Watts–Strogatz) to heterogeneous ones (here modified Holme–Kim model). Our team is currently investigating these differences and details, to have a clearer picture of how those affect complex contagions. For instance, the concept of transitivity (global clustering coefficient) diverges from the one of local clustering coefficient, in conjunction with the nontrivial impact of the heterogeneous degree distribution. For these elements to be properly studied and dealt with, we would need a full manuscript, which is currently under preparation (as a sequel to the present manuscript).

2) I'm far from mastering the subject but, basically, it is quite easy to understand this work once you have precise definitions of the quantities and models considered. About this, I found it frustrating to be forced to look at the references and at the methods section in order to get a precise idea of the work done by the authors.

I believe that something can be put in the main body such as the update rule for LTM and LFC and the definitions of polarization speed and collective frequency response. Sure, the authors can keep the definitions of the network metrics in the method section but I would spend some words to remember the known results on WS networks.

We agree and understand the Reviewer's frustration. The issue boils down to the fact that the Methods section arrives a bit late in the text. We have added the definition of the polarization speed earlier in the next. However, we found it challenging to add the dynamics of the LTM and LFC (Eqs. (1) and (2)) in the Results section. Indeed, these equations require the introduction of numerous secondary quantities, which would therefore have to be mathematically introduced and defined. This would create a significant duplication in content within the manuscript.

3) What do authors expect from an ‘‘underdamped’’ dynamics governed by a second order ODE of LFC? I mean the simple model of harmonic oscillator obtained by substituting the second derivative of x_i to the left side of eq. 2) and, eventually, by adding a damping term too. In a way (continuous limit with local interaction) we are moving from the heat equation (with damped and diffusive information propagation) to the wave equation (with linear and no damped information propagation) and

therefore the two models have clearly different behaviors. Can we expect something new and/or different in the simple/complex contagion scenario?

I believe that a few words on the subject can help to make the results even more general and interesting. Moreover, the collective frequency response function is obtained analytically in a simple way in this case also, so it doesn't seem complicated to have some numerical results by exploiting the network metrics already computed by the authors on the LFC model. However, it is a request without 'penalty', I only ask the authors to think about it, to give me their opinion and, eventually, to add something in the paper if they are reasonably convinced to improve it.

This is an interesting question and it appears very much connected to the analysis by Cavagna and collaborators (Ref. [37]: Cavagna, A., Giardina, I., & Grigera, T. S. (2018). The physics of flocking: Correlation as a compass from experiments to theory. *Physics Reports*, **728**, 1–62.).

From the perspective of Sec. 3.3.2 in Ref. [37], our first-order in time LFC (Eq. (2)) is unable to sustain propagating waves even if behavioral cascades can take place. The idea of an underdamped second-order dynamics has been introduced by Cavagna and collaborators in Ref. [36], which helped explain the fast pace of information transfer among flocking birds performing a rapid collective turn. Hence, as suggested by Reviewer #2, it would be of interest to consider a second-order leader-follower consensus dynamics that includes a "moderate" damping term.

First, it is worth mentioning that although there is a vast body of research on second-order consensus, the literature on second-order leader-follower consensus is fairly limited and primarily focused on controlling the behavior (i.e. the state variable of the agents). To the best of our knowledge, only Naomi Leonard and collaborators (Punzo, G., Young, G. F., Macdonald, M., & Leonard, N. E. (2016). Using network dynamical influence to drive consensus. *Scientific Reports*, **6**(1), 1-13.) have looked into the patterns of emergent collective behaviors. In this article, the authors looked primarily into a first-order consensus, but also considered its extension to a second-order consensus—a consensus behavior associated with the agent's positions (hence the second-order dynamics from Newton's second law). Specifically, a double consensus is considered: consensus on relative positions and velocity. Interestingly, some underdamped oscillatory behaviors are obtained (see Fig. 6). However, these collective responses are obtained by driving the leader in ways that are essentially akin to a step response. As explained below (see response to Reviewer #3 – item 2), we have found that the study of a collective step response with a first-order LFC fails to discriminate between the notable differences in behavioral propagation (simple vs. complex contagion) at different time-scales.

In conclusion, the promising preliminary results of an underdamped collective response reported by N. Leonard and collaborators could be extended to a collective frequency response—such as the one reported in the present manuscript—in line with the analysis by Cavagna and collaborators. This idea is now stated in the Discussion (6th Paragraph, lines 313-316 in main, 384-387 in diff).

Finally, in my opinion, minor changes only are necessary in order to deserve the publication on Nature Communications.

We appreciate that Reviewer #2 feels that only minor changes are necessary.

Comments by Reviewer #3

Reviewer #3 (Remarks to the Author):

The manuscript argues that complex contagion is a general feature of models of consensus spreading, not limited to threshold-based models with binary options. Instead, a transition or crossover from simple to complex contagion is observed also in models with continuous state variables, although it is strongly influenced by network topology and the intrinsic frequency of the spreading behaviour.

To reach this conclusion the authors study three cases. The first is the linear threshold model (LTM) on the Watts-Strogatz small-world network. This is a model with binary options where a site can flip from state 0 to state 1 if a fraction of its neighbours larger than a threshold θ is in state 1. The network is initialised in state 0 except for one site and its neighbours which are set to 1, and the spread of state 1 across the network is monitored. The responsiveness of the system is quantified through the polarisation speed (which measures how long it takes for a given fraction of the network to switch to state 1), and this is measured for networks with average degree $k=16$ and various rewiring probabilities so that they have different clustering coefficient (C), average shortest path (l) and Kirchoff index (R_g). It is observed that for low thresholds the speed increases with decreasing l , C and R_g , while at large threshold this trend no longer holds (indeed, it is reversed for some intervals of l , C , and R_g). This is quantified using Spearman's rank correlation coefficient, which changes sign from negative at low thresholds to positive at large thresholds. This inversion is interpreted as a transition from simple to complex contagion (Fig 1).

The second case is the leader-following consensus (LFC) model, where the individual site's state is given by a real number $x_i(t)$. A leader executes an arbitrary trajectory $u(t)$, and the rest of the nodes follow it with a dynamics given by a first-order ordinary differential equation (eq 2), which reduces to a forced overdamped harmonic oscillator for the case of a single follower. The model is studied again on the small-world network. Here the metric for the network responsiveness is the linear response $H^2(\omega)$. A simple-to-complex transition is found where the Spearman rank correlation shows that at low frequencies response is higher for lower l and R_g (with no correlation to C), while at high frequencies response is higher for higher l and C (and low, but still negative, correlation with R_g). Here networks of different average degree are studied so that the three metrics (l , C , R_g) are not tied to each other.

The third model (which is both simulated and actually implemented in hardware) is a swarm of ten robots which can tune their orientation heading so that it aligns with their neighbours. One robot is chosen as leader and is made to turn its heading with constant frequency, while the rest follow their neighbours with a nonlinear law (eq 12). Three different interaction networks are studied. The response is quantified with a different response function (eq 13). The response is found to decrease with increasing frequency for all networks, but while at low frequencies the response is the same for all networks, there is a range of frequencies where the network with the highest C has a clearly larger response. Higher frequencies cannot be studied experimentally, but simulations show that at high frequencies all networks again have the same response.

In my opinion the manuscript needs significant improvement to make a stronger case for the conclusion, and to be clearer in its message. The main weaknesses I find in the manuscript are:

We would like to thank Reviewer #3 for the in-depth reading of our manuscript and for the detailed summary of our work. We believe that the extensive revision undertaken addresses all the comments and recommendations of Reviewer #3 (see below).

1. The definition of complex contagion is not clear. In the introduction the authors write that in simple contagion "behavior propagates through a single exposure or interaction", while complex contagion needs "social reinforcement", which is explained as transmission that "requires an individual to have contact

with two or more sources of activation’’. One would expect a more quantitative definition of complex contagion, at least one that is applicable to the models considered here. Or at least, a discussion of how this qualitative definition translates to the metrics that will be later employed to detect the nature of contagion. It is surprising to read in the discussion (p9 1304) that ‘‘one is lacking a definite way of characterizing the behavioral contagion’’. This seems to cast doubt on the conclusions the authors are drawing! This phrase suggests that there are conceptual problems related to the very definition of complex contagion. This is fine: the present results may help resolve these problems. But given that complex contagion is the central element of the manuscript, these issues should be dealt with up-front and made clear from the beginning.

Reviewer #3 touches on a very important point, and we fully agree with her/him on this particular issue of a lack of quantitative definition of a complex contagion. Simple contagions are well-known and well defined. On the other hand, the issue with the definition of complex contagions has been around for a long time. As a matter of fact, authors working on this topic deal in many different ways with this lack of a mathematically articulated definition of complex contagions (e.g., Rosenthal et. al. (2015) PNAS, Centola (2018) Princeton University Press, Törnberg (2018) PLOS ONE, Centola & Macy (2007) Am. J. Sociol., O’sullivan et. al. (2015) Frontiers in Physics, Iacopini et. al. (2019) Nat. Comms. and Refs. [3,4,13,20,44,64]). Furthermore in Guilbeault & Centola (2021) Nat. Comms., the authors still use this same exact definition. Although complex contagions have been around for 14 years, it seems that the original definition from Centola still prevails. This explains why we quoted it verbatim (like many other researchers do (Refs. [3,4,13,20,44,65])). Even if the original definition by Centola & Macy is broad and lacks a quantitative/mathematical form, it has the merit to exist and stress the key general features behind a complex contagion. In our opinion, the two main issues with this definition are the following. First, and as already mentioned, it lacks a mathematical/quantitative articulation, and therefore it is not straightforward to use. But most important of all, this original definition is closely tied to Centola & Macy’s original modeling framework, namely the LTM., where the latter has been the sole modeling framework considered to study complex contagions. It might be possible that this definition has had an unexpectedly limiting effect on the research on complex contagion. In connection with our present work, it was necessary to start from the existing and prevailing definition (which again, is not ours but is the one commonly accepted by the scientific community). Reviewer #3 finds the definition of a complex contagion not clear but, as a matter of fact, it is the only one currently available and widely used (Refs. [3,4,13,20,44,65])).

Although, our aim with this work is not to provide a new and more specific definition of complex contagion, we do provide a systematic way of characterizing a complex contagion. We hope that this characterization would offer the research community new ways to study and identify complex contagions in other fields of study.

Reviewer #3 rightly points to our statement (On line 338 in main, 353 in diff): ‘‘one is lacking a definite way of characterizing the behavioral contagion’’. We indeed mention that because we believe that our work addresses this issue and provides a systematic way to overcome the issues with the existing prevailing definition. To make this point more salient and to make a stronger case for our contribution, we have added/edited the following statements in the Introduction:

‘‘These threshold models fit perfectly Centola & Macy’s original definition of a complex contagion, whereby a transition from simple to complex contagion takes place when increasing the threshold beyond a value corresponding to having more than one activated neighbor [13,16,17]. However, the key concept of activation may not be as straightforward when considering models lacking a threshold, which can become an issue when trying to use the existing definition of a complex contagion.’’

Reviewer #3 is also right saying that our ‘‘present results help to resolve these problems’’. Following her/his advice, we included that up front in the Introduction:

‘‘Using a new way of characterizing complex contagions, we uncover their existence in consensus-based dynamics. Specifically, we shed a new light on some ...’’

2. The authors argue that a transition from simple to complex contagion is probably a general feature of a broad class of collective decision-making processes. But

the simple-to-complex contagion transition is studied in three models using a different metric for each model. The case would be stronger if the same metric were shown for all models. For example, it seems that the polarisation speed concept can be generalised to the LFC: One starts with all sites at $x=0$, fixes the leader at $u=10$ (say) and counts the number of sites that have exceeded some threshold value. Or a linear response similar to that employed for the LFC could be used for the robots. If this is not possible due to technical reasons, these reasons should be made explicit, and some argument given as to why the metrics chosen can be regarded as equivalent.

With this comment, Reviewer #3 touches on yet another important point, namely the connection between our collective decision-making models (simulations with the LTM, theory with the LFC, and robotic experiments with the nonlinear heading consensus). Reviewer #3 rightfully acknowledges that with all three models, we identify the nature of the behavioral propagation taking place—simple or complex contagion—by probing the variations of a given metric with the network topology. Given the fundamentally different constructs of the three models, a single common metric is unlikely to exist. However, we agree with Reviewer #3 that if a given metric allows us to reveal the very nature of a behavioral propagation, it must capture some key features of the collective response. This is a technical point that we investigated during our study but chose not to include in our original submission. But, we agree with Reviewer #3 that these technical details connecting the three metrics help make our analysis more general.

These technical elements are detailed below along with a new paragraph added to the manuscript. Let us first stress some technical aspects:

- By construction, the LTM is a discrete-time protocol with discrete (binary) state values. It is worth recalling that in the LTM (linear threshold model), the linearity refers to the threshold and not to the dynamics. The presence of a threshold introduces an additional binary component to this collective decision-making protocol.
- By construction, the LFC is a linear continuous-time protocol with continuous state values devoid of any threshold, binary and/or discrete component.
- As for the nonlinear heading consensus experiment, it entails a protocol with time-sampled continuous state values that are governed by a nonlinear dynamics.

With all three models, our interest lies with how large swaths of the system respond to the propagation of a particular behavior, and accordingly we study the response of these systems to induced changes in some of the state values. The origin of these state changes can either be endogenous or exogenous depending on the problem studied. For instance, when considering a school fish threatened by a predator, the change in state is exogenous.

- For the LTM: behavioral changes are triggered endogenously by initially seeding a fraction of nodes: a particular node and its direct neighbors in the network sense. Indeed, given its particular construction, a behavioral propagation can only be induced by changing the state of more than one agent. This constraint therefore prevents us from considering a single leader agent with the LTM. As a next step, we analyze and measure how fast this change in behavior propagates through the system. This is quantitatively obtained by means of the polarization speed. Moreover, given the discrete-time nature of the LTM, it is natural to consider a **step response**, i.e., a response to a sudden change (at $t = 0$) in some state variables (our seed nodes).
- For the LFC and the nonlinear heading consensus: the behavioral change is obviously triggered by the variations of the leader's state; we are dealing with a leader-follower consensus after all. Given the lack of threshold, the controlled variations of the leader's state (by means of its frequency ω) are sufficient to yield a behavioral propagation throughout the group, which is measured by means of the collective frequency response. With both the LFC and the nonlinear heading consensus, it is natural to consider a **frequency response**, which informs us about how the behavioral propagation occurs for different values of the frequency ω of the leader's state variable. Nonetheless, our interest remains with the actual responsiveness of the system to the injected behavior. Although, we could have considered a transient response (in the form of a step response or impulse response), we have found that the (collective) **frequency response** is a more appropriate metric to analyze

this responsiveness toward an injected behavior as it classically disentangles the effect of each and every frequency present in the step function.

As is well-known in system dynamics and control theory, the derivative of the (unit) step response of a given system yields the (unit) impulse response, whose Fourier transform is the frequency response. Like the impulse function, the unit step function contains all frequencies in the spectrum albeit with a spectrum decaying in $1/\omega$. Mathematically, the step response and the frequency response provide the same information and insight about the system's response. In practice, and depending on the nature of the system, both the step response and frequency response can be equivalently considered although one response maybe more easily implementable than another. It should be added that although the step response and the frequency response yield the same mathematical information, the frequency response has the advantage of informing us about the exact dynamics of each independent harmonic, whereas the step response yields a continuous sum of all the harmonics. This particular point is particularly important when considering systems having very distinct dynamics at different time scales (i.e., for different frequencies). Obviously, implementing the step response with the LTM is straightforward while the frequency response clearly is not. With the LFC and the nonlinear heading consensus, both responses have been considered but only the frequency response is presented in our manuscript. Contrary to our results with the frequency response shown in the manuscript, those with the step response are much harder to interpret. Interestingly, this challenge with the step response can easily be understood from our results with the frequency response (Fig. 2). When considering the step response, the low-frequency harmonics are expected to propagate according to a simple contagion while the higher frequency harmonics follow the pattern of a complex contagion. This linear combination of harmonics in the step function clearly hinders our goal to distinguish a simple contagion from a complex one.

Beyond those mathematical considerations, it is also worth looking at this question from the modeling standpoint. For instance, if we consider a flock of birds attacked by a very slow predator, this would be equivalent to a low frequency in which the system has plenty of time to respond. Conversely, when considering the typically fast predatory actions, the natural system has to contend with a high frequency, and a high collective frequency response at that particular frequency is indicative of high responsiveness, which is what is necessary for the group survival. In this example from nature, this would translate into a sharp collective turn, and such an evasive maneuver is the outcome of a rapid collective decision-making achieved by the flock in response to the behavioral response to the birds that spotted the predator.

In summary, the collective system dynamics and the associated metrics studied in this manuscript are theoretically connected. The polarization speed and collective frequency response both measure the responsiveness of the system but to different input signals: a step response and a frequency response respectively. However, the distinctive features of the LTM and the LFC—(i) discrete-time vs. continuous-time, (ii) with threshold vs. without threshold, (iii) response to multiple state changes vs. response to a single state changes—conceptually prevent the use of a single identical quantitative metric. For instance, a single leader would have no effect whatsoever on the LTM dynamics. Even a group of leaders (say 30%) in the LTM would not be able to operate at different frequencies given the binary nature of their states.

3. The swarm experiment is presented as a third scenario where complex contagion is found. Although it seems that this case is quite different from the first two (LTM and LFC models), the robots interact on a fixed network, so that this ‘‘swarm’’ is actually a sort of LFC but with a nonlinear contagion law (eq 12 vs eq 2). A key characteristic of swarms is that as individuals move, the interaction network changes (since the interactions are local); this crucial ingredient is one of the reasons the study of such systems is so challenging. In the present case, the positions of the robots play no role: neighbouring robots (in the network sense) interact only through their orientation (heading), eq 12. In an actual swarm, orientation variations lead to variations of position that in turn can cause changes in the interaction network. This coupling between positions, orientations and interaction network is absent in the model studied, and in this sense the term ‘‘swarm’’ is misleading. ‘‘Robot network’’ would be a more appropriate term.

The emphasis with our experiments is to verify that a complex contagion can occur with a nonlinear consensus law, therefore we kept the three network topologies static. Given this assumption of static

networks, we agree with Reviewer #3 that the term “swarm robotic” could be somehow misleading. As a consequence, when referring to our testbed, we have replaced the term “swarm” by either “multi-robot systems” or “networked robots”.

4. The robot results do not seem to show simple-to-complex transition, at least in the way this transition was shown in the first two models. Instead of an inversion of the trends with C , l , R_g , the response is either independent of the network or has a definite trend. Also, it is not possible to tell which of the network metrics is the main factor influencing the response.

Reviewer #3 is absolutely right. We highlighted in our manuscript that our aim with these experiments was to provide evidence of a complex contagion. This is why the section dealing with the networked robotic experiments does not delve into the simple-to-complex transition. It is centered on the complex contagion phenomenology.

It is worth mentioning that simple contagions have been widely studied with multi-robot systems (e.g.: “Ultrafast consensus in small-world networks” by R. Olfati-Saber (2005)) and the network distance has been recognized as the key parameter driving simple contagions. As a matter of fact, our experiments do not allow us to conclude whether the Kirchhoff index R_g is a better network-distance indicator than the classical average shortest path l . But this particular question has been addressed with a larger IoT testbed in the following reference:

Nikolaj Horsevad, “Understanding Collective Behaviors Through Network Topology”, Ph.D. Thesis (2021) <http://dx.doi.org/10.13140/RG.2.2.29489.51044>

Other issues/comments:

1. The simple-to-complex transition is studied in networks of fixed size (10000 nodes for the LTM, 240 nodes for the LFC). Networks of different sizes would be needed to have a hint on whether e.g. the right panel of fig 1 would show a sharper or less sharp transition for larger networks; similarly for fig 2. Also, fig S1 shows that the threshold value at which the transition happens is significantly affected by the choice of cascade size: $\theta=0.2$ is complex at 10% cascade size, but simple beyond 50%. Here also studying different network sizes would help clarify whether the situation tends to be sharper for larger networks.

This is a valid point and we have run additional computations with different network sizes. Specifically we now have the following new results/figures:

- LTM with $N = 5,000$ agents/nodes: results shown in the S.I. (new Fig. S2) and below (Fig. 1). One can notice that with this smaller system size, the results remain qualitatively unchanged.
- LFC with $N = 120$ agents/nodes: results shown in the S.I. (new Fig. S5) and below (Fig. 2). One can also notice here that with this smaller system size the same inversion of the trend is observed, and it is only less marked than with the larger system.

These additional results with smaller system sizes confirm the trends analyzed and discussed in our original submission. As anticipated by Reviewer #3, larger system sizes only offer more markedly apparent trends.

2. In fig 1, systems that do not reach 30% contagion are excluded. How does that affect the results? In fig S1, the effect of changing the 30% threshold to other values is studied, but it is not clear why the red curve always stops at $\theta=0.3$. In the caption of fig 1 it is stated that some lines stop prematurely because the 30% cascade size is never reached; one would then expect that changing the cascade size would also change the point at which the lines must stop.

This is an excellent observation of the peculiar stop of the red curve in Fig. 1 and Fig. S1 (and now also visible in the new Fig. S2) at the particular threshold value $\theta = 0.3$. First, it is worth noting that this red curve corresponds to the lowest clustering coefficient value considered ($C = 0.34$). It is well known

that networks with such low clustering are unable to sustain the global spread of a complex contagion with a high threshold in the LTM. Specifically, with these small-world networks at low clustering (i.e., practically random networks), there is either systematic local-spreading leading to a global cascade (at low threshold values) or no local spreading and therefore no global cascade at all (at high threshold). When looking carefully at the data obtained with our LTM for different cascade sizes, we observed that for $\theta = 0.3$ and $C = 0.34$, there is practically no local spreading, hence no ensuing global cascade. When increasing the clustering coefficient (“less random network”), the system experiences some level of local spreading, which yields macroscopic cascades, that are not reaching the entire system. As noted in our manuscript, past studies have focused on global cascades, where 100% of the nodes are activated, whereas here we considered partial, yet macroscopic cascades. This fact is clearly visible when comparing the high-threshold trends in Fig. S1 (a) (smallest partial cascade of 10%) versus Fig. S1 (i) (largest partial cascade of 90%). In summary, this observation is clearly a consequence of the very nature of the LTM given its binary/threshold features.

3. The units of $\nu(t)$ (polarisation speed) are, according to eq 1, the inverse of the time unit, rather than length divided by time. The name ‘‘polarisation rate’’ is thus perhaps more appropriate for this quantity.

Semantically speaking, Reviewer #3 is absolutely right. However, the term “speed” is the most commonly used when referring to spread/contagion/propagation on networks. Hence, to be in line with the literature dealing with spreading on networks, we prefer to keep the term “speed”.

4. It is stated that the only parameter of WS networks is the rewiring parameter p . In fact, the average connectivity is also a parameter that controls network topology, which is actually also changed by the authors when studying the LFC model. The authors explain that it is necessary to study networks of different degrees to untangle the effect on contagion of the different network metrics C , l , R_g . The question is why this is done only for the LFC and not for the LTM.

Reviewer #3 is right. Strictly speaking, to define a class of WS networks, one has to select the number of nodes, the average degree and a value for p . However, to facilitate the study, we kept the number of nodes and average degree constant, while only varying the rewiring parameter p .

However, one could argue that such an analysis of the influence of the average degree on the contagion processes with the LTM although certainly interesting, would not provide any additional result contributing to our study. Indeed, our analysis of the LTM is an extension of the work by Centola (Centola & Macy (2007) *Am. J. Sociol.*) and Duncan Watts (Watts, (2002) *PNAS*) (Refs. [13,17]), whereby we focused on the influence of the clustering, average shortest path and Kirchhoff index, thus keeping the average degree constant and varying p . Since our analysis uses the known results and mathematical derivations, we did not feel the need to further study the influence of the average degree on the contagion process.

With the LFC—our primary focus—no prior study is available and changing the average degree became a necessity to disentangle the various network effects.

5. The expression ‘‘pace of the intrinsic behavior’’ used in the abstract is not very clear to me. Do the authors mean frequency of the leader movement?

Reviewer #3 is right in saying that the pace of the intrinsic behavior is equivalent to the frequency of the leader. This point is related to the main point 2. above, in which we explain that the frequency response allows one to assess the propagation of behavior with different time scales. Indeed, when a particular behavior with a given pace, its Fourier transform will yield a specific frequency. Hence by probing the collective (frequency) response at any frequency, we can directly deduce how a given behavior with a given pace will propagate.

The details added to the manuscript in response to point 2. above should make this point clearer as well.

REVIEWERS' COMMENTS

Reviewer #1 (Remarks to the Author):

The authors addressed all my comments, and I am happy to support publishing.

Reviewer #2 (Remarks to the Author):

I have carefully read the revised draft and the authors' responses to all criticisms from the referees. I find that they have done a great job, very careful and precise, so I have no doubts in recommending publication in Nature Communications.

Reviewer #3 (Remarks to the Author):

The questions raised in my first review have been satisfactorily answered by the authors for the most part. Unfortunately not much of the detailed response to my comments is reflected in the new version. At this point I am convinced of the technical soundness of the work, though I still have some doubts on the presentation.

In my opinion, the manuscript is hard to follow unless one is quite familiar with the jargon and previous references discussing complex contagion. As pointed out by reviewer #2, it would help if the models and metrics were introduced before the results. The authors acknowledge that "methods section arrives a bit late", it is not clear to me whether editorial requirements prevent the authors to reorganise the manuscript in a way that is more helpful to the reader, since the changes in response to the reviewers' comments are mostly local.

In summary, the authors' responses have cleared up the technical points, but the revised manuscript style still seems tailored for a more specialised journal.

Transition from simple to complex contagion in collective decision-making

NCOMMS-21-23105-T

Nikolaj Horsevad, David Mateo, Robert E. Kooij, Alain Barrat, Roland Bouffanais

Note: all edits made to the original version of the manuscript are highlighted in blue for additions and in red for deleted items in the text appended at the end of this document.

Comments by Reviewer #1

Reviewer #1 (Remarks to the Author):

The authors addressed all my comments, and I am happy to support publishing.

We would like to thank Reviewer #1 for supporting publication at this stage.

Comments by Reviewer #2

Reviewer #2 (Remarks to the Author):

I have carefully read the revised draft and the authors' responses to all criticisms from the referees.

I find that they have done a great job, very careful and precise, so I have no doubts in recommending publication in Nature Communications.

We would like to thank Reviewer #2 for the encouraging remarks and for supporting publication.

Comments by Reviewer #3

Reviewer #3 (Remarks to the Author):

The questions raised in my first review have been satisfactorily answered by the authors for the most part. Unfortunately not much of the detailed response to my comments is reflected in the new version. At this point I am convinced of the technical soundness of the work, though I still have some doubts on the presentation.

We appreciate Reviewer #3 confirming that we have adequately answered her/his comments and that she/he is confident about the technical soundness of our work. We are glad to be given the chance to further improve the presentation of our results.

In my opinion, the manuscript is hard to follow unless one is quite familiar with the jargon and previous references discussing complex contagion. As pointed out by reviewer #2, it would help if the models and metrics were introduced before the results. The authors acknowledge that "methods section arrives a bit late", it is not clear to me whether editorial requirements prevent the authors to reorganise the manuscript in a way that is more helpful to the reader, since the changes in response to the reviewers' comments are mostly local.

We understand that Reviewer #3 recommends bringing forward the Methods section, since it contains all the technical details regarding the models and metrics used (LTM, LFC). Reviewer #3 asked if the location of the Methods is imposed by any editorial requirements. Following the in-principle approval of our manuscript, we received a detailed Editorial Checklist set by the Editorial Office. It turns out that the Method section has to stay in its current location, after the Introduction, Results and Discussion sections.

To alleviate this editorial constraint, and to address Reviewer #3's concerns, we have now added some technical details about the models and metrics prior to them being used in the Results section. Below is the list of edits and items that are relevant to this round of revision:

- Previously available technical details: The definition of complex contagion was given verbatim in the Introduction (lines 23-27). More details and key knowledge about complex contagions were also given in the 3rd paragraph of the Introduction.
- Line 89 (Line 89 below): we have added some details about the particular network metrics that are considered in our study.
- Between lines 111 and 112 (Between lines 112 and 113 below): We have now added the explicit mathematical formulation (Equation 1) of the LTM at the very beginning of the Results section. These details and mathematical equations are new (not moved from the Methods section).
- Line 123 (Line 124 below): The mathematical expression of the polarization speed was added during the second round of review and should help the reader.
- Lines 138-139 (Lines 139-140 below): We gave the explicit mathematical definition of the Kirchhoff index R_g and we stressed its connection with the eigenvalues of the Laplacian matrix.
- Between lines 177 and 178 (Between lines 178 and 179 below): we have now added the explicit mathematical formulation (Equation 2) of the LFC in the Results section, prior to its use. The reader is still referred to the Methods section for additional technical details.

All these details are highlighted in the annotated manuscript appended at the end of this response letter.

In summary, the authors' responses have cleared up the technical points, but the revised manuscript style still seems tailored for a more specialised journal.

We believe that this new round of edits to our manuscript addressing all recommendations listed in the previous point will make this manuscript more easily accessible to readers not versed into the field of complex contagion.